 TOOLS AND RESOURCES

# Subcellular sequencing of single neurons reveals the dendritic transcriptome of GABAergic interneurons

**Julio D Perez[1], Susanne tom Dieck[1], Beatriz Alvarez-Castelao[2], Georgi Tushev[1], Ivy CW Chan[3], Erin M Schuman[1]\***

[1]Max Planck Institute for Brain Research, Frankfurt am Main, Germany; [2]Department of Biochemistry and Molecular Biology, Veterinary School, Complutense University of Madrid, Madrid, Spain; [3]Department of Behavior and Brain Organization, Center of Advanced European Studies and Research, Bonn, Germany

**Abstract** Although mRNAs are localized in the processes of excitatory neurons, it is still unclear whether interneurons also localize a large population of mRNAs. In addition, the variability in the localized mRNA population within and between cell types is unknown. Here we describe the unbiased transcriptomic characterization of the subcellular compartments of hundreds of single neurons. We separately profiled the dendritic and somatic transcriptomes of individual rat hippocampal neurons and investigated mRNA abundances in the soma and dendrites of single glutamatergic and GABAergic neurons. We found that, like their excitatory counterparts, interneurons contain a rich repertoire of ~4000 mRNAs. We observed more cell type-specific features among somatic transcriptomes than their associated dendritic transcriptomes. Finally, using celltype-specific metabolic labeling of isolated neurites, we demonstrated that the processes of glutamatergic and, notably, GABAergic neurons were capable of local translation, suggesting mRNA localization and local translation are general properties of neurons.

**\*For correspondence:**
erin.schuman@brain.mpg.de

**Competing interests:** The authors declare that no competing interests exist.

## Introduction

The synaptic connections between the dendritic and axonal processes of individual neurons provide the substrate for the flow, modification, and storage of information in the brain. Because they usually extend hundreds of micrometers from their cell bodies, axons and dendrites must operate with some biochemical and electrical autonomy to accommodate changes at the speed of synaptic transmission. This is accomplished by post-translational modifications and regulation of the local proteome, including the local synthesis of proteins. By localizing specific mRNAs to these compartments, a neuron can control where and when different proteins are made and function (*Holt et al., 2019*). Working with tissue-derived RNA samples, multiple groups have identified transcripts present in dendrites and axons, revealing which proteins are likely generated by local translation (*Cajigas et al., 2012*; *Glock et al., 2020*; *Gumy et al., 2011*; *Hafner et al., 2019*; *Poon et al., 2006*; *Zhong et al., 2006*). However, by design, these tissue-based approaches forgo fundamental features of local transcriptomes that can only be dissected at the level of individual neurons. First, how universal is the localization and translation of mRNAs in different neuronal cell types? Second, how variable are the dendritic/axonal transcriptomes from one neuron type to another? Third, how much influence does the somatic abundance of a given mRNA have on its localization in neuronal processes?

It is unclear if subcellular mRNA localization and translation occurs in all types of neurons, and whether it varies according to cell type or cell state. The current state-of-the-art includes two kinds of sequencing studies. On one hand, bulk RNA studies of microdissected tissue (containing multiple

cell types) have profiled mRNAs in the neurites (*Poon et al., 2006*) or brain slice neuropils (*Cajigas et al., 2012*; *Glock et al., 2020*; *Zhong et al., 2006*) providing a comprehensive view of mRNAs localized to the axonal/dendritic population. However, quantitative expression values obtained from these bulk sequencing approaches do not reflect the actual mRNA values present in an individual neuron, but rather the population average, potentially hiding intercellular heterogeneity in the local transcriptome. On the other hand, unbiased single cell RNA-seq (scRNA-seq) studies circumvent the averaging effect of bulk approaches and have revealed extensive diversity in cellular composition of the brain. However, these methods require the generation of cell suspensions, resulting in the shearing off of neuronal processes and the profiling of exclusively somatic mRNAs (*Prakadan et al., 2017*). In addition, most bulk-sequencing studies have attributed the identified mRNAs to excitatory neurons; it is still unknown whether local translation occurs in different classes of inhibitory interneurons. Indeed a better understanding of the subcellular transcriptome of these neurons may help explain their diverse size, transmitter phenotype, axonal and dendritic architecture, and electrophysiological properties (*Huang and Paul, 2019*; *Pelkey et al., 2017*). Moreover, our growing appreciation of the brain's cell type complexity and the fact that many of the variable genes act at the synapse (*Saunders et al., 2018*; *Zeisel et al., 2018*) suggest that the specialization of dendritic and axonal compartments is a critical consequence of cell differentiation. Determining which mRNAs are locally translated could thus highlight unique functions and dynamics of cell type-specific compartments.

Single cell approaches that include dendrites or axons could, in principle, elucidate how different classes of neurons (glutamatergic, GABAergic, peptidergic, etc.) establish and maintain diverse transcriptomes across different compartments. Since it houses the source of transcription (the nucleus), the neuronal soma is the source of mRNA from which local transcriptomes are generated. It is unclear, however, how much a transcript's somatic level influences its localization in processes, and whether transcripts can be enriched or excluded from the neurites regardless of their somatic abundances. In fact, mRNA localization is often defined based on either the absolute abundance of a transcript in neurites or the relative enrichment in comparison to somata (*Kügelgen and Chekulaeva, 2020*). These absolute versus relative values can generate significantly different abundance rankings of the mRNAs that comprise local transcriptomes. For example, a highly expressed somatic transcript that is also abundant in dendrites would be considered highly dendritic in absolute terms, but less so in relative terms. The opposite could be true for a lowly expressed transcript frequently found in dendrites albeit at low copy numbers. These distinctions could be biologically meaningful: absolute values may indicate the number of molecules needed in a compartment (*Kosik, 2016*), while relative values may reflect regulation of mRNA concentrations between compartments (*Liu et al., 2016*). Estimates of these values based on bulk expression may be distorted since diversity among individual neurons in both the somatic and local transcriptomes is averaged out. Ideally, the transcriptome of each subcellular compartment would be measured and compared within each neuron. Such analyses could also elucidate the functions and thus the biochemical environments that distinguish each neuronal compartment.

Here we developed a method to profile, in an unbiased manner, the transcriptome of subcellular compartments of an individual neuron, which can be scaled to hundreds of single cells. We used this method to identify the dendritic and somatic transcriptomes of cultured hippocampal neurons and observed mRNAs in the dendrites of all types of neurons including three GABAergic types. We found that dendritic transcriptomes were different from somatic transcriptomes and detected distinct patterns of dendritic mRNA variability according to cell type. Additionally, we describe the relation between somatic abundance and dendritic localization; we found that whether an mRNA is dendritically enriched or de-enriched strongly correlates with its cellular function. Finally, we show that proteins are locally synthesized in the dendrites of both glutamatergic and GABAergic neurons, suggesting mRNA localization and local translation are general neuronal properties.

## Results

### Development of a subcellular scRNA-seq method

To profile individually the somatic and dendritic transcriptome of a single neuron we used laser capture microdissection (LCM), previously used to profile the transcriptome of individual cells from

tissue sections (*Foley et al., 2019*; *Nichterwitz et al., 2016*), since it allows for the dissection of biological material with μm resolution. To target subcellular structures with LCM, we first optimized the dissection of individual dendritic processes of rat primary cultured hippocampal neurons (see Materials and methods). To maximize sensitivity and throughput, we adapted a scRNA-seq protocol for droplet microfluidics (*Macosko et al., 2015*) which tags mRNAs with both an index and unique molecular identifier (UMI). The index allows sample pooling and processing followed by the identification and quantification of mRNAs from individual neuronal somata and dendritic arbors (*Figure 1A*). Using 92 synthetic RNA standards of different concentrations, we determined the method accurately measures RNA expression levels and reliably detects RNA species present in as low as four copies per sample (*Figure 1—figure supplement 1A and B*).

We thus generated a large data set of dendritic arbors and somata from single neurons which were relatively isolated within a culture dish. Every neuron was first imaged, followed by capture of the soma, followed by collection of all accessible dendrites (*Figure 1—figure supplement 1C*). Neurites of smaller diameters were avoided since they could be axons. We also collected additional somata from other neurons to expand the depth of cell types explored. As a negative control, we laser-captured comparable areas within the dish with no visible cellular structure ('empty cuts'). All samples were then processed as described above (also see Materials and methods). Data processing steps and analyses of the sequencing data are summarized in *Figure 1—figure supplement 1D*. In total, we collected 795 samples: 276 dendritic samples and their respective somata, 227 somata without a dendritic arbor, and 16 empty cuts.

Somatic and dendritic samples contained significantly more RNA molecules and unique genes than empty cuts (*Figure 1B*; *Figure 1—figure supplement 1E*, mean RNA counts = somata 64,265, dendrites 6,802, empty cuts 41), highlighting the accuracy of our laser-capture microdissection. On average we detected 3404 unique gene features in somata, 553 features in dendrites, and 19 features in empty cuts. Some cellular samples exhibited very low counts (somata and dendritic samples with less than 700 and 55 genes detected, respectively) and were thus not retained for downstream analyses (*Figure 1—figure supplement 1E*). We analyzed the depth of our subcellular transcriptomes and estimated that we identified the majority of genes expressed in the soma, whereas for dendrites, the most abundant dendritic genes were identified, but some less abundant genes were probably below our detection limit (*Figure 1—figure supplement 1E*). To identify potential glia or low-quality cells we performed uniform manifold approximation and projection (UMAP) for all samples, and identified two clusters expressing either glial or apoptotic markers (*Figure 1—figure supplement 1F and G*); these were removed from further analyses. After the above filtering steps, 345 somatic and 112 dendritic samples remained.

## Somatic and dendritic samples exhibit distinct transcriptomic signatures

We first investigated the similarities between somatic and dendritic samples. If dendrites simply reflect somatic expression we would expect them to group together with their respective soma in a transcriptome-based clustering analysis. As seen by UMAP visualization, however, cellular samples clearly separated according to subcellular compartments: somata and dendrites (*Figure 1C*; *Figure 1—figure supplement 2A*). Somata were further differentiated between GABAergic and glutamatergic types, while dendrites remained as one cluster. To determine whether differences in transcriptome depth (*Figure 1B*; *Figure 1—figure supplement 1E*) were responsible for the observed clustering, we randomly downsampled somata to different molecular totals, comparable to those observed in dendrites. Although the differences between compartments and cell types became less explicit, dendrites still formed a separate cluster while somata continued to split according to cell type (*Figure 1—figure supplement 2B*). We also compared our data set with a recent publication (*Middleton et al., 2019*), which used a different method to profile the dendrites and soma of 16 single neurons; we found that the dendrites and somata of the Middleton et al. data set integrated with the dendrites and somata of our data set (*Figure 1—figure supplement 2C*). Together these observations suggest that intercellular diversity is less pronounced in dendrites than in somata, but do not rule out cell type-specific effects in the dendritic transcriptome. Additionally, they imply that significant differences exist between the dendritic and somatic transcriptomes of the same neuron.

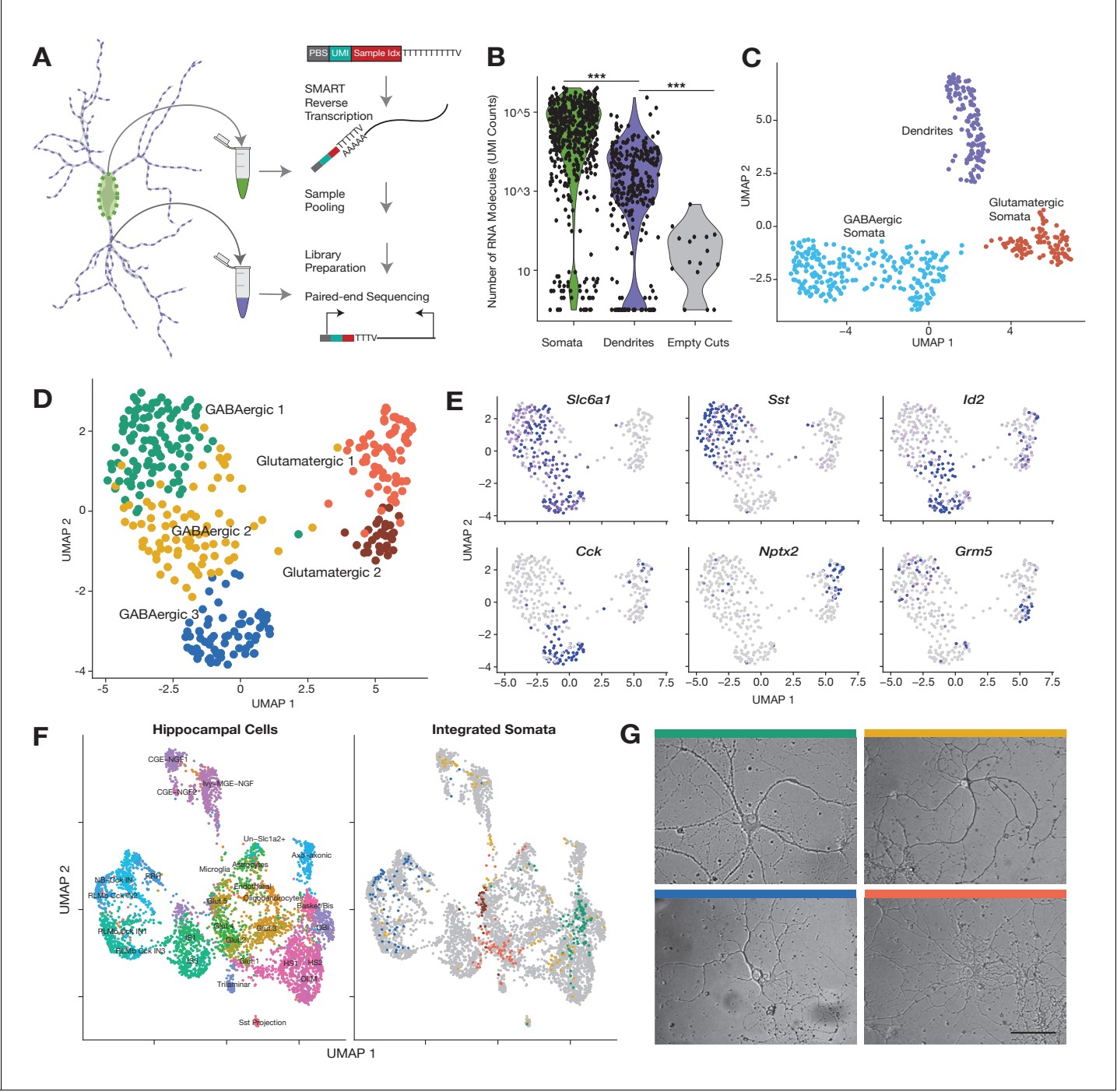

**Figure 1.** Subcellular single cell RNA-seq (scRNA-seq), compartment-specific transcriptomic signatures, and identification of cell types based on the somatic transcriptome. (A) Experimental workflow: After fixation, the soma and dendritic arbor of individual cultured hippocampal neurons were acquired separately by laser capture microdissection. After membrane lysis, mRNAs from a single soma or dendritic arbor were reverse transcribed using a primer containing a PCR primer binding site (PBS), unique molecular identifier (UMI), and a sample index. A template switch reaction added a complementary PBS at the mRNA 5' end. Samples were then pooled, amplified, and prepared for paired-end sequencing. See also Materials and methods and *Figure 1—figure supplement 1D* for a description of the sequencing data analysis. (B) Number of RNA molecules (UMI counts) detected in different sample groups. (C) Uniform manifold approximation and projection (UMAP) plot of somatic and dendritic samples, colored by cluster identity. (D) UMAP plot of somata samples, colored by cluster identity, revealed two glutamatergic types and three GABAergic cell types. (E) Normalized expression of marker genes projected onto UMAP plot in D. (F) Seurat integration was used to find cell type correspondence between hippocampal tissue single neurons (from two separate studies *Zeisel et al., 2015* and *Harris et al., 2018*) and single somata acquired in this study. Left panel shows the cluster organization of hippocampal tissue cells only, and right panel shows the integration of single somata from this study

*Figure 1 continued on next page*

Figure 1 continued

(color indicates cell type as established in D) with hippocampal tissue cells (gray). For left panel: *Basket/Bis*: Basket and bistratified cells, *CGE-NGF*: Caudal ganglionic eminence neurogliaform, *Glut*: Glutamatergic, *HS*: Hippocampo-septal interneuron, *IS*: Interneuron-selective interneuron, *Ivy-MGE-NGF*: Ivy medial ganglionic eminence neurogliaform, *NB*: Cck+ non-border interneuron, *OBi*: Oriens-bistratified neurons, *OLM*: Oriens-lacunosum-moleculare, *Sst+ projection* neurons, *RLMb*: Cck+ Radiatum-lacunosum molecular border interneuron, *RRH*: Radiatum-Retrohippocampal neurons. (G) Images of representative cells (that were subsequently harvested using laser capture) exhibiting morphology most commonly observed among identified cell types. Color of bar over image indicates cell type as indicated in D. Scale bar = 75 µm.

The online version of this article includes the following figure supplement(s) for figure 1:

**Figure supplement 1.** Subcellular single cell RNA-seq (scRNA-seq), genes detected, and compartment-specific transcriptomic signatures.
**Figure supplement 2.** Compartment-specific transcriptomic signatures.
**Figure supplement 3.** Somatic inhibitory marker expression, integration of single cell RNA-seq (scRNA-seq) data sets, and morphological analyses.

## Identification of cell types based on the somatic transcriptome

To investigate the effect of cell type on the dendritic transcriptome, we first characterized the cell types present in our data set using our somatic samples. Dimensionality reduction of the somata revealed five neuron types: two glutamatergic and three GABAergic (*Figure 1D*). *Figure 1—figure supplement 2D* shows the most differentially expressed genes between identified cell types (adjusted p<0.05, logistic regression model). The major differences in expression occurred between glutamatergic and GABAergic somata; however, significant differences between subtypes were also observed. Although glutamatergic types 1 and 2 largely co-expressed a majority of transcripts, a handful of mRNAs like *Nptx2* and *Grm5* allowed their distinction (*Figure 1E*). GABAergic 1 neurons, the most abundant cell type in our data set, were distinguished by their expression of *Sst* and other neuropeptides. Despite their clear inhibitory identity, GABAergic 3 neurons exhibited a very different transcriptomic profile from GABAergic 1, mainly distinguished by the expression of *Cck* and *Cnr1*. Finally, GABAergic 2 neurons showed similarities in their expression patterns to both GABAergic 1 and 3; their identity was best described by the co-expression of some GABAergic 3 markers, like *Id2*, but not others, like *Cck* (*Figure 1E*; *Figure 1—figure supplement 2D*). Within the entire GABAergic population, we also observed distinct expression patterns of *Pvalb*, *Vip*, and *Reln* and other interneuron markers within or across the cell types (*Figure 1—figure supplement 3A*), suggesting these clusters harbor further cell type diversity.

To benchmark the breadth of cell types detected we compared our cell types to those previously described in two CA1-derived scRNA-seq data sets, which profiled the transcriptome of 1314 CA1 cells (*Zeisel et al., 2015*) or 3663 CA1 GABAergic neurons (*Harris et al., 2018*). Our own clustering analysis of this data set identified 12 or 19 cell types, respectively (data not shown), consistent with the authors' observations. To find transcriptomic similarities that were robust to different species (rat vs. mouse), different growth conditions (primary culture vs. tissue), and different developmental stages (cells cultured from P0 brains vs. juvenile/adult brains), we used the Seurat.v3 algorithm (*Butler et al., 2018*; *Mayer et al., 2018*). Integration of the two tissue data sets and our somata samples organized cells into multiple glutamatergic, GABAergic, and non-neuronal cell types (*Figure 1F*; *Figure 1—figure supplement 3B*). Our glutamatergic 1 somata mainly distributed between hippocampal glutamatergic 1 and 2 neurons, while glutamatergic 2 somata overwhelmingly integrated within the hippocampal glutamatergic 5 cluster. Among the shared marker genes between tissue and cultured cells were *Schip1* and *Nrgn* for glutamatergic 1 somata, and *Camk2a* and *Grm5* for glutamatergic 2 somata. GABAergic 1, 2, and 3 somata showed clear differences in their integration patterns. GABAergic 1 somata were mainly split according to the expression of *Pvalb* among other genes. *Pvalb*+ GABA1 somata integrated among Basket-bistratified interneurons, while *Pvalb*- GABA1 somata integrated among hippocampo-septal and oriens-bistratified neurons. On the other hand, GABAergic 3 somata integrated exclusively in *Cck*+ neighborhoods, particularly onto radiatum-lacunosum moleculare border interneurons (RLMbs). Finally, GABAergic 2 somata were split between different types of neurogliaform and HS neurons.

Cell type classification based on the somatic transcriptome also correlated with differences in cell morphology. *Figure 1G* shows representative examples of each cell type. GABAergic neurons possessed darker cell bodies than glutamatergic neurons (*Figure 1—figure supplement 3C*) consistent with previous observations (*Benson et al., 1994*). Among GABAergic cells, we noted that

GABAergic 1 and GABAergic 3 neurons tended to have the largest and smallest somata, respectively, and exhibited differences in the length and thickness of their dendrites (*Figure 1—figure supplement 3D,E*). These morphological features were consistent with the transcriptomic relations established in *Figure 1F*. The transcriptome of GABAergic 1 most resemble basket-bistratified and hippocampo-septal interneurons: large cells that have their somata in the stratum pyramidale of the hippocampus, and exhibit complex dendritic arborizations that extend to other layers (*Pelkey et al., 2017*). On the other hand, the transcriptome of GABAergic 3 is closest to RLMb interneurons: smaller cells, whose somata are located at the border between the stratum radiatum and the stratum lacunosum moleculare, and exhibit short dendrites that usually remain in the same layer. Together these results reveal some clear cell type identities within hippocampal cultures that recapitulate the transcriptome signatures and resemble the morphological patterns observed in tissue.

## Cell type-specific effects: the dendritic transcriptome of GABAergic interneurons

Profiles of the neuronal local transcriptome have been derived from populations containing diverse cell types or from populations enriched with glutamatergic neurons. Instead, our single cell approach allowed us to determine the local mRNA pool of cells classified according to their (somatic) transcriptome, including GABAergic types. We detected ~4000 mRNA species in the dendrites of the two glutamatergic and three GABAergic neuron types (*Figure 2A*; *Supplementary file 3*). This number is consistent with the number of translated mRNAs recently observed in the hippocampal neuropil (*Glock et al., 2020*) and represents 30% of the transcripts detected in somata of each cell type, suggesting similar patterns of mRNA localization in glutamatergic and different GABAergic types (*Figure 2B*; *Supplementary file 3*). The dendritic transcriptome of GABAergic neurons exhibited both similarities and differences to the glutamatergic dendritic transcriptome (*Figure 2A*). For instance, *Calm1* and *Map1a* were among the most abundant dendritic RNA across all cell types. We also observed similarly low abundances in the dendrites of all cell types for genes like *Meg3* and *Psmb2*, which are robustly expressed in the somata of all cell types (*Figure 2B*). Conversely, we observed mRNAs whose dendritic abundances changed according to cell type including *Lgi2* and *Serf2*, which were most abundant in GABAergic 1 dendrites, and *Cnr1* and *Cplx2* which were most abundant in GABAergic 3 dendrites. GABAergic 2 neurons exhibited comparable levels of dendritic mRNA species to the other cell types, but did not appear to preferentially localize any mRNA species. To validate the presence of mRNAs in GABAergic dendrites with an independent method, we performed two-color single molecule FISH (smFISH) for selected candidates in cells expressing *Gad1/2* (*Figure 2C*; *Figure 2—figure supplement 1A*). As predicted from the scRNA-seq data, *Atp5f1b*, *Cox8a*, and *Fth1* mRNAs were all detected in proximal and distal dendrites, while *Psmb2* mRNA was mostly absent from dendrites. Quantification of puncta in *Gad*+ and *Gad*− distal dendrites for the above transcripts and *Camk2a* (known to localized to glutamatergic dendrites) confirmed these observations (*Figure 2D*; *Figure 2—figure supplement 1B*). *Camk2a* was abundant in the distal dendrites of *Gad*− but not *Gad*+ neurons indicating *Gad*− neurons were mostly glutamatergic. On the other hand, *Atp5f1b*, *Cox8a*, and *Fth1* were abundant in the distal dendrites of both *Gad*+ and *Gad*− neurons, while *Psmb2* was absent from the dendrites of both neuron types. Taken together, these results indicate that GABAergic neurons also localize thousands of mRNAs to their dendrites, resulting in both similarities and differences to the glutamatergic dendritic transcriptome.

To determine statistically significant variability in the dendritic transcriptome according to cell type, we tested differential expression (DE, see Materials and methods for a full description) in dendrites according to cell type for the following comparisons (*Figure 2—figure supplement 1E*): glutamatergic vs GABAergic, GABAergic 1 vs GABAergic 2–3, and GABAergic 2 vs GABAergic 3. The same comparisons were also made between somata. Consistent with unsupervised clustering results above (*Figure 1C*), substantially fewer genes exhibited significant DE between dendrites than between somata (*Figure 3A*; *Supplementary file 4*). Nevertheless, valuable information could be extracted from the differentially expressed genes. Consistent with the observed transcriptomic distances among cell types (*Figure 1D*), more DE genes were detected between glutamatergic and GABAergic dendrites than in either of the GABAergic comparisons. Interestingly, we observed that in each comparison some genes differentially expressed in dendrites were also differentially expressed in somata and vice versa (*Figure 3A and B*; *Figure 3—figure supplement 1A*). For example, *Hpcal4*, *Cort*, and *Cnr1* were expressed in a cell type-specific manner in both

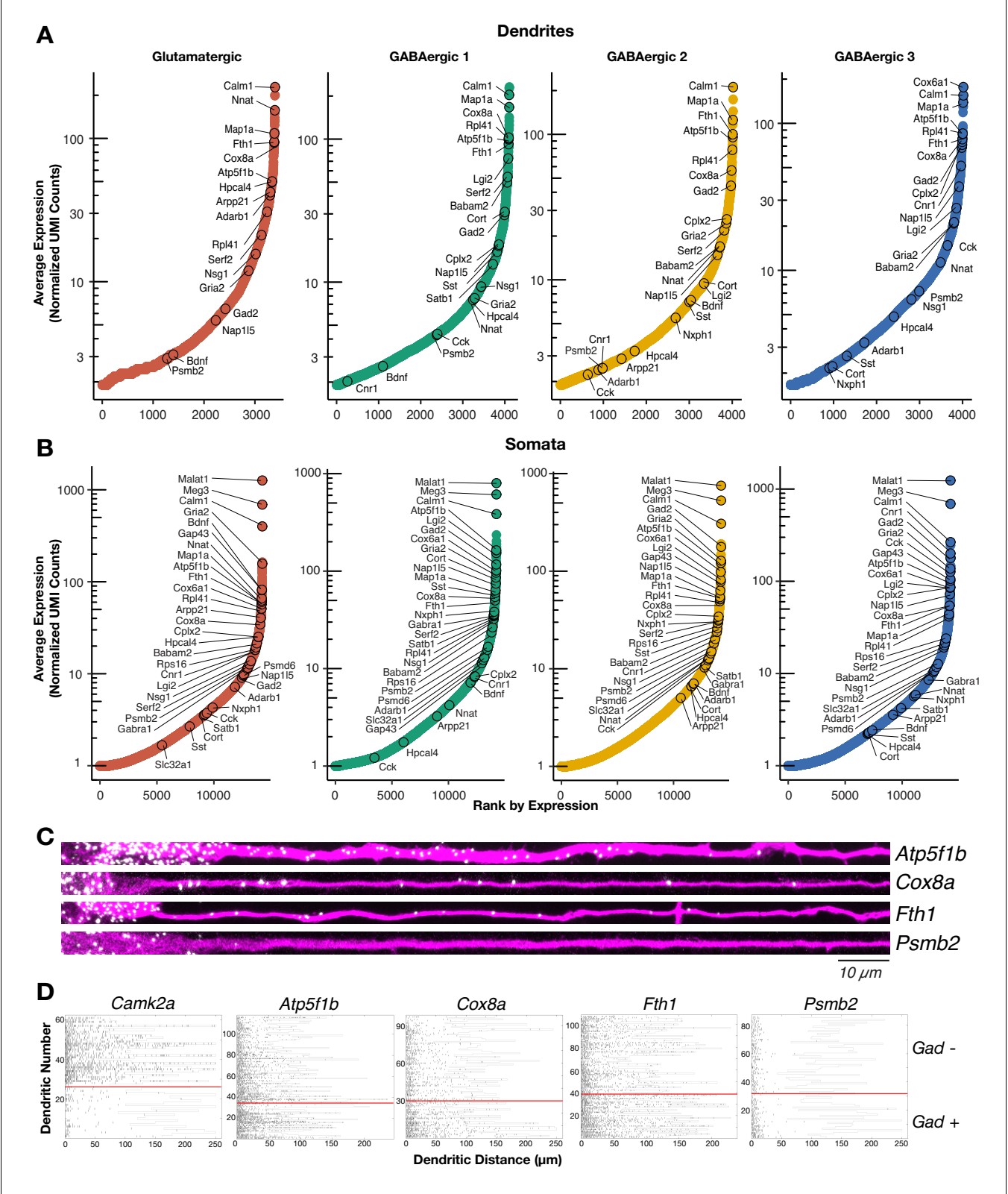

**Figure 2.** The dendritic transcriptome of GABAergic interneurons. (**A**) Plots showing the average normalized unique molecular identifier (UMI) counts of genes detected in the dendrites (≥1.9 molecule per sample on average) of the indicated cell types. X-axis shows genes ranked according to their expression from lowest to highest. Some genes are indicated by name. N = glutamatergic (11), GABAergic 1 (36), GABAergic 2 (30), and GABAergic 3 (18). (**B**) Same as **A** but for somatic samples. N = glutamatergic (95), GABAergic 1 (103), GABAergic 2 (82), and GABAergic 3 (65). (**C**) smFISH for

*Figure 2 continued on next page*

*Figure 2 continued*

indicated mRNAs (in white) observed in GABAergic dendrites immunolabeled with an anti-Map2 antibody (magenta). GABAergic identity was determined by smFISH *Gad1/2* as seen in *Figure 2—figure supplement 1A*. Scale bar = 10 µm. (D) Raster plot showing smFISH dendritic detection pattern over a large number of dendrites. Dendrites of both excitatory and inhibitory cells were straightened from the same images, sorted according to the expression of *Gad1/2* (positive below red line, negative above red line) and the gene of interest channel displayed, after automated peak detection, as a raster plot.

The online version of this article includes the following figure supplement(s) for figure 2:

**Figure supplement 1.** Cell type-specific effects in the dendritic transcriptome.

somata and dendrites. However, not all cell type-specific genes present in somata were detected in dendrites, for example, *Bdnf*, *Satb1*, and *Adarb2*. Curiously, we also observed transcripts preferentially detected in the dendrites of specific cell types even though they were not differentially expressed among somata, suggesting that post-transcriptional mechanisms might act in a cell type-specific manner to regulate mRNA localization. *Adarb1*, *Babam2*, and *Cyc1* are examples of this dendritic pattern.

To validate the above findings with an independent method we performed two-color smFISH for selected candidates, namely *Hpcal4*, *Bdnf*, and *Cnr1* (*Figure 3—figure supplement 1B*) together with the cell type marker mRNAs *Gad1/2* or *Cck*. Consistent with the sequencing data, *Hpcal4* and *Bdnf* were expressed in *Gad*-negative neurons, while high *Cnr1* coincided with *Cck* expression (*Figure 3—figure supplement 1B,C*). Also, we found that although both *Hpcal4* and *Bdnf* were expressed preferentially in *Gad*-negative neurons at similar levels, *Hpcal4* signal was detected at higher levels in dendrites (*Figure 3C*), as observed in the sequencing data. To assess the variability between neurons we analyzed a larger set of dendrites for these two candidate genes and clustered them with respect to their *Gad* signal. As expected, *Gad*-positive cells exhibited almost no signal for either mRNA while *Gad*-negative neurons clearly showed widespread dendritic signal for *Hpcal4* but not for *Bdnf* (*Figure 3D*). Together, these data suggest that cell identity can regulate the dendritic mRNA population and that, in addition to cell type-specific transcription, downstream mechanisms may dictate the dendritic localization of specific mRNAs depending on the cell type.

## Relation between the somatic and dendritic transcriptomes of single neurons

It is unknown how much the somatic abundance of an mRNA influences its subsequent localization, or not, to the dendrites. To investigate this, we focused on 95 of the 112 neurons for which both the somata and corresponding dendrites passed QC filters. This set contained neurons from all cell identities and thus the following analyses do not differentiate by cell type. We observed that, like the somatic transcriptome, dendritic mRNA abundances roughly resembled a log-normal distribution although it was more zero-shifted (*Figure 4—figure supplement 1A*). This indicated that the majority of dendritic mRNAs were detected at relatively low levels while a few transcripts were found at moderate-to-high levels. Using a logistic regression model, we observed that somatic expression significantly influenced dendritic detection (*Figure 4A*, p=$1.29 \times 10^{-6}$), but only partially explained the observed variance (McFadden Pseudo $R^2$ = 0.33). To compare the somatic and dendritic transcriptome, we used normalized expression values that stabilize technical variability among samples for genes present in more than 25% of samples (*Hafemeister and Satija, 2019*). Somatic and dendritic expressions were significantly correlated (*Figure 4B*, generalized linear regression model, p=$2 \times 10^{-16}$) but, again, only partially accounted for the observed variance ($R^2$ = 0.50). These results indicate that the more abundant an mRNA is in the soma, the more likely it is to be present and abundant in dendrites, but additional mechanisms also act to increase or decrease dendritic presence of certain mRNAs.

To identify genes with significant enrichment or de-enrichment in dendrites we performed a paired-differential expression analysis using a Poisson generalized linear model (*Stuart et al., 2019*), based on the difference between somatic and dendritic values in each single neuron. We used normalized values that correct for the large differences in detected molecules between somatic and dendritic samples (*Figure 1B*), and thus, allowed us to test for relative enrichment (*Hafemeister and Satija, 2019*). We detected 463 and 764 genes enriched or de-enriched in dendrites, respectively

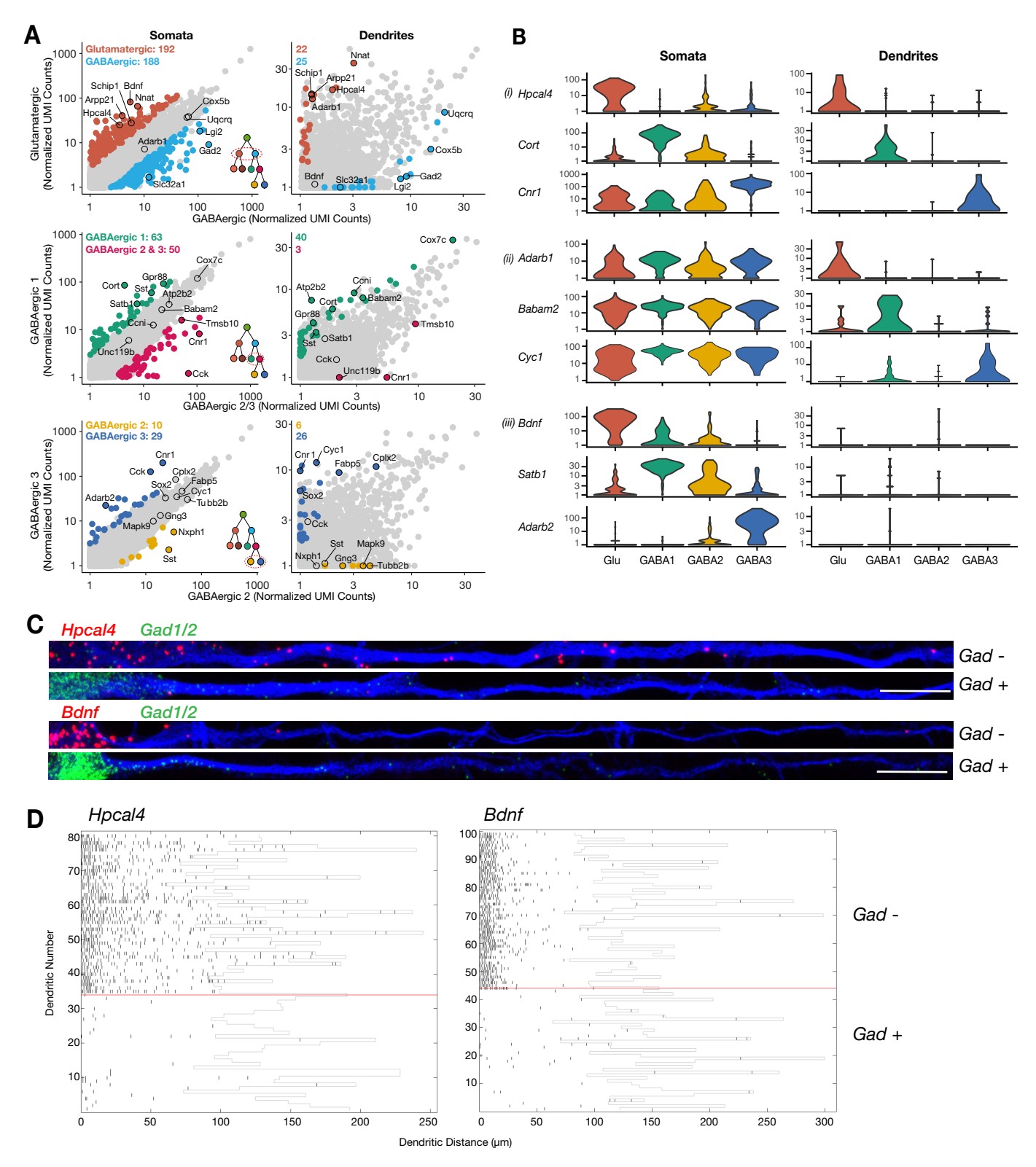

**Figure 3.** Cell type-specific effects in the dendritic transcriptome. (**A**) Scatterplots comparing mean expression between somata or dendrites of different cell types as indicated by x- and y-axes. Significant differentially expressed genes (logistic regression model) are colored in each panel; some significant genes are indicated by name. Inserts in the bottom right of somata scatterplot indicate the comparison within the established hierarchy of cell types (see *Figure 2—figure supplement 1E*), and on the top left corner of each panel the number of significant genes for each group. Somata N =

*Figure 3 continued on next page*

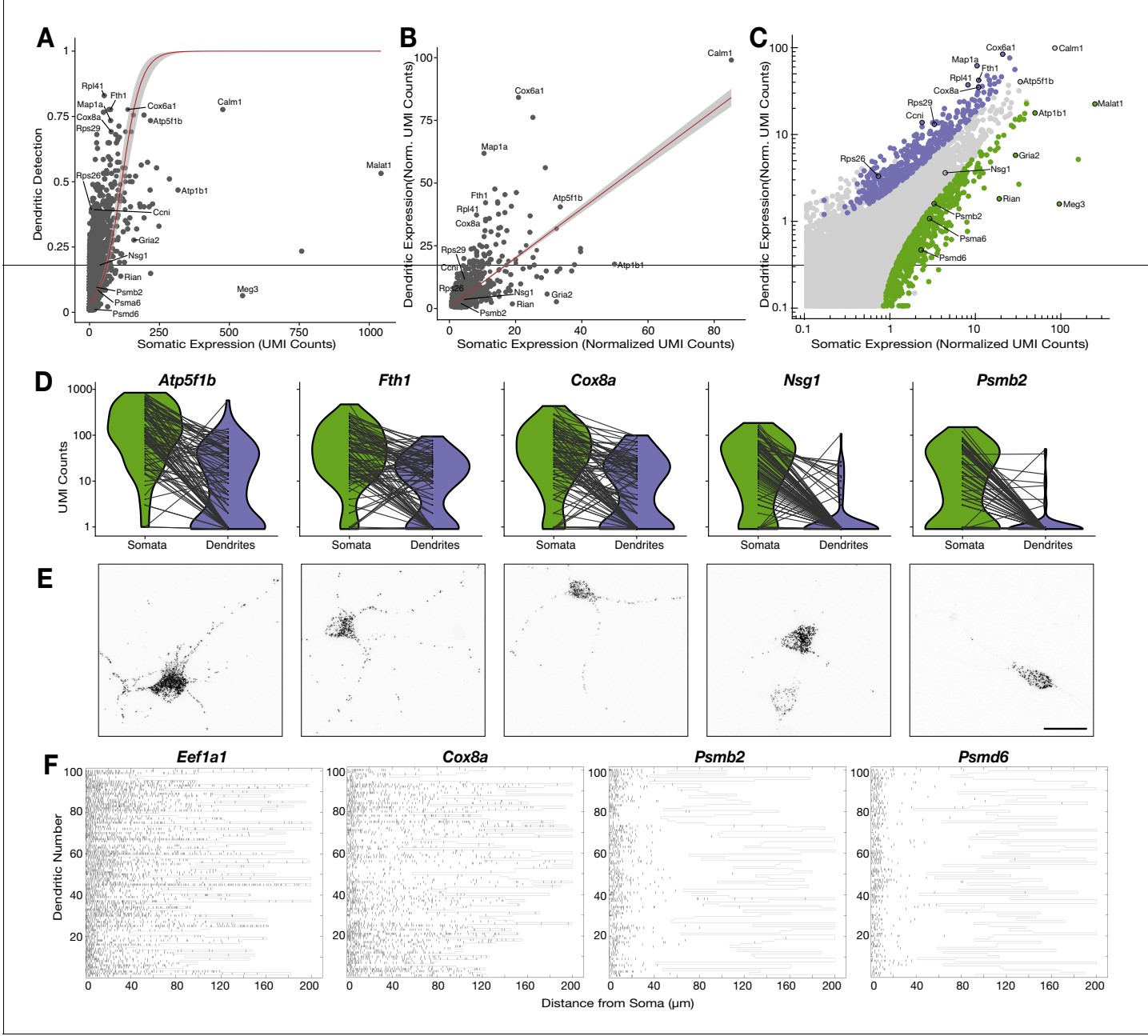

**Figure 4.** Relationship between the somatic and dendritic transcriptomes of single neurons. (A) Scatterplot showing the relationship between somatic abundance (in average unique molecular identifier [UMI] counts) and the fraction of samples in which an mRNA species is detected in the dendrites. A logistic regression model revealed a significant correlation (p=1.29 × $10^{-6}$, McFadden's Pseudo $R^2$ = 0.33). Gray area indicates the 95% confidence interval. Some genes discussed in the text are indicated by name. (B) Correlation between somatic and dendritic expression for genes expressed in at least 20% of dendritic samples. Red curve shows a linear regression model (p=2 × $10^{-16}$, $R^2$ = 0.50). Some genes discussed in the text are indicated by name. Gray area indicates the 95% confidence interval. (C) Scatterplot showing the results of paired differential expression analysis using a Poisson generalized linear model. Colored dots indicate statistically significant genes (adjusted p<0.05) in somata (green) or dendrites (purple); some significant genes are indicated by name. (D) Violin plots showing the raw counts in somata and dendrites of gene examples. Lines between violins indicate the paired values of soma and dendrites from the same neuron. (E) FISH validations for genes indicated in D. The abundance and distribution of mRNAs detected (black) are shown. Neurons were identified by anti-Map2 immunolabeling (see images in *Figure 4—figure supplement 1C*). Scale bar = 25 µm. (F) Raster plot showing the cell-to cell variability of mRNA localization as determined by smFISH, for candidate dendritically enriched (*Eef1a1*, *Cox8a*) and dendritically de-enriched (*Psmb2*, *Psmd6*) mRNAs. One dendrite was chosen per cell, straightened, and the profile converted via peak detection into a raster plot (displayed from soma/proximal dendrite to distal dendrite, from left to right).

The online version of this article includes the following figure supplement(s) for figure 4:

*Figure 4 continued on next page*

*Figure 4 continued*

**Figure supplement 1.** Relationship between the somatic and dendritic transcriptomes of single neurons.

(*Figure 4C*; *Supplementary file 5*), including most of the genes deviating from the general trends in *Figure 4A and B*. *Figure 4D* and *Figure 4—figure supplement 1B* depict transcripts using their raw counts and with compartments paired according to cell, to more accurately present the quantitative nature and variability of individual neurons.

Beside relative enrichment, localization of an mRNA in dendrites may be evaluated based on its absolute local abundance. Indeed, we observed significant enrichment in some, but not all, transcripts with high abundance in dendrites. For instance, the mitochondrial genes *Atp5f1b*, *Fth1*, and *Cox8a* all had high absolute values in dendrites, but only the latter two were significantly enriched (*Figure 4D*). This is because a lower fraction of the cell's total *Atp5f1b* mRNA is localized to dendrites in comparison to *Fth1* and *Cox8a*. We also observed significant dendritic enrichment and de-enrichment across a wide spectrum of absolute values. For example, the mRNAs of ribosomal proteins *Rps26*, *Rps29*, and *Rpl41* were all comparably enriched in dendrites although they had very different dendritic expression values (*Figure 4—figure supplement 1B*). On the other hand, *Atp1b1*, *Malat1*, and *Gria2* were all de-enriched in dendrites even though they had moderate-to-high abundances in both dendrites and somata (*Figure 4—figure supplement 1B*). In contrast, genes like *Meg3*, *Rian*, *Psmb2*, and *Psmd6* were similarly de-enriched and scarce in dendrites, although they are expressed at very different levels (*Figure 4—figure supplement 1B*). Since *Malat1*, *Meg3*, and *Rian* are nuclear non-coding RNAs their dendritic de-enrichment was to be expected. However, the strong dendritic de-enrichment of genes such as *Gria2*, *Psmb2*, and *Psmd6* suggests that specific mRNAs may be actively deterred from entering dendrites.

To validate these observations, we performed smFISH on five representative genes (*Figure 4E*; *Figure 4—figure supplement 1C*). Consistent with our scRNA-seq data set, *Atp5f1b*, *Fth1*, and *Cox8a* exhibited similar high abundances throughout the length of dendrites. However, as predicted, *Atp5f1b* was more abundant in the soma than *Fth1* and *Cox8a*. For *Nsg1*, a gene that was neither significantly enriched nor significantly de-enriched in dendritic samples, the great majority of the signal was in the soma, with some signal detected in the proximal dendrites. Finally, as seen in our scRNA-seq data, signal for *Psmb2* was almost exclusively present in the soma. To characterize the cell-to-cell variability of such patterns we used smFISH to quantify dendritic localization of two dendritically enriched genes, *Eef1a1* and *Cox8a*, and two dendritically de-enriched genes, *Psmb2* and *Psmd6*, across hundreds of neurons (see Materials and methods; *Figure 4F*; *Figure 4—figure supplement 1D*). Equivalent lengths of dendrites were measured for each gene (*Figure 4—figure supplement 1E*). Both *Eef1a1* and *Cox8a* were highly abundant, often reaching far into the dendrites of most neurons. In contrast, *Psmb2* and *Psmd6* displayed a rapid decay of signal density within the first 20 μm of the dendrite. We also observed some cell-to-cell variability in the absolute number of dendritic mRNAs of a given transcript, as also seen in our scRNA-seq data (*Figure 4D* and *Figure 4—figure supplement 1B*). For instance, a handful of dendrites exhibited low expression for *Cox8a*, while for *Psmb2* and *Psmd6* a few neurons showed sparse mRNA puncta along the dendrites. Thus, as observed by scRNA-seq data, the smFISH experiments showed that different genes substantially differ in their dendritic localization and abundance, with some cell-to-cell variability not accounted for by cell type.

## Functional associations of dendritically enriched and de-enriched genes

To investigate whether mRNA distribution between soma and dendrites is related to function, we performed a rank-based gene set enrichment analysis. Genes were ranked by a 'representation score', which is the product of their mean expression × detection frequency, and tested for enrichment in gene ontology cellular and synaptic components (GOCC and SynGO; *Koopmans et al., 2019*) terms. Most terms were strongly represented in both somata and dendrites, and included essential dendritic functions like energy and mitochondria, cytoskeleton, and synaptic organization and maintenance (*Figure 5A*; *Supplementary file 6*). Although their local function is still unclear, ribosomal protein mRNAs were also highly represented in dendrites (*Cajigas et al., 2012*; *Glock et al., 2020*; *Gumy et al., 2011*; *Poon et al., 2006*; *Zhong et al., 2006*). Interestingly, we

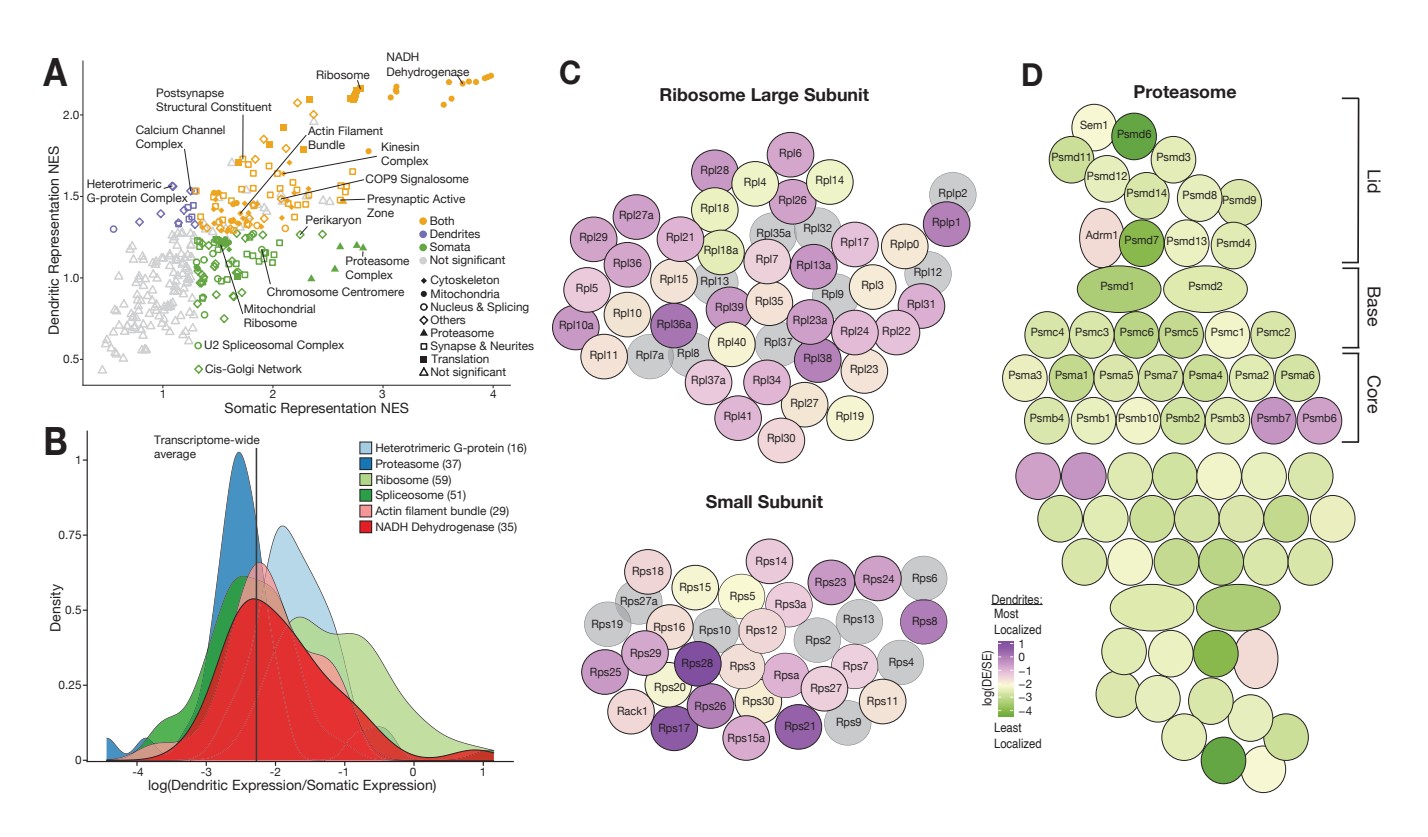

**Figure 5.** Functional associations of dendritically enriched and de-enriched transcripts. (A) Gene ontology analyses of dendritic enrichments. Shown are the normalized enrichments scores (NES) of different cellular component and synaptic terms (GOCC and SynGO) calculated from ranked-lists of dendritic and somatic mRNAs. Genes were ranked according to a 'representation score' which equals detection frequency × mean expression. Terms with a significant enrichment score (FDR < 0.25) are colored according to their significant compartment. Some terms are indicated by name. (B) Density plot showing the distribution of the relative dendritic enrichments for genes belonging to six GOCC terms. Relative enrichments equated the average ratio between dendritic and somatic expression in a single neuron. Distributions are colored according to the complex and the number of genes evaluated for each complex is shown in parenthesis next to the term in the figure legend. (C) Cartoon showing the proteins of the ribosome colored according to the relative dendritic enrichments of their mRNAs. Gray color indicates that the mRNA species was not detected. (D) Same as C but for proteins of the proteasome.

also found a set of genes exclusively and significantly enriched in dendrites related to autophagy, calcium channels, and heterotrimeric G-protein complexes. As predicted by *Figure 4*, we also found a set of terms de-enriched in dendrites including chromosome regulation, splicing, and Golgi-related terms, whose proteins are known to be excluded/diminished in dendrites (*Hanus et al., 2016*). Intriguingly, proteasome-related terms are also de-enriched in the dendritic transcriptome even though proteasomal proteins are present and play essential roles in dendrites (*Bingol and Schuman, 2006*). Thus, the local transcriptome supplies dendrites with some, but not all, of the proteins needed on-site.

We next determined whether the mRNAs encoding proteins that are integral members of macro-molecular complexes exhibited similar or differential enrichment in the two compartments. We selected GOCC terms describing macromolecular complexes and analyzed the relative enrichment distribution of the mRNAs in the somata vs. dendrites. *Figure 5B* shows the distribution of six different complexes. The distributions of both heterotrimeric G-protein complexes and ribosomal proteins exhibited an above-average dendritic enrichment (*Figure 5B and C*), but the variance was higher in the latter. Both the proteasome and spliceosome were skewed toward below-average dendritic enrichment (*Figure 5B and D*), with the proteasome exhibiting lower variance. These observations suggest that protein function influences mRNA localization patterns.

## Local protein synthesis occurs in the dendrites of GABAergic neurons

As described above, we found that GABAergic neurons localize thousands of mRNAs to their dendrites (*Figure 2A*), including both cell type variable and invariable transcripts, suggesting that these neurons also have the capacity to locally synthesize proteins. To test this directly, we took advantage of a conditional mouse line that enables the expression of a mutant Methionyl tRNA synthetase (MetRS*) in specific cell types, when crossed to different Cre driver lines (*Alvarez-Castelao et al., 2017*). Coupled with the administration of the non-canonical amino acid ANL (azido norleucine), protein synthesis was measured in glutamatergic neurons (Nex-Cre::GFP-2A-MetRS*), GABAergic neurons (Gad2-Cre::GFP-2A-MetRS*), and somatostatin-expressing neurons (Sst-Cre::GFP-2A-MetRS*). We cultured cortical neurons from these mouse lines on membranes that enabled the separation of neurites from soma-containing fractions (*Alvarez-Castelao et al., 2020*; *Poon et al., 2006*), and examined protein synthesis in intact neurons ('full labeling', ANL added to entire dish) or neurites alone ('neurite only', ANL added after somata layer was removed) using BONCAT (*Alvarez-Castelao et al., 2019*; *Dieterich et al., 2007*; *Figure 6A*). As expected, full labeling resulted in signal both in the soma and the neurites (*Figure 6B and C*). Following 'neurite only' labeling, strong signal was detected in the neurites of all cell types examined, including the GABAergic neurons (*Figure 6B and C*). These results confirm that GABAergic neuron dendrites locally synthesize proteins and suggest that this is a pan-neuronal property.

## Discussion

We established a method to characterize the transcriptomes of subcellular compartments (cell bodies and dendrites) in individual neurons which we used to measure both the intercellular variability and intracellular allocation of the dendritic transcriptome. This technique allows for the clean dissection of the dendrites and respective soma from a cell of choice, while preserving information about its original morphology. Compared to previous attempts at profiling the local transcriptome of single neurons (*Middleton et al., 2019*; *Tóth et al., 2018*), our technique dramatically expands the number of cells profiled and mRNAs identified. We discovered that the dendrites of GABAergic neurons contain ~4000 mRNA species, similar to the number of mRNAs observed in glutamatergic dendrites. We found that the dendritic transcriptome varied according to cell type, but to a lesser extent than the somatic transcriptome. Additionally, we described the relation between the somatic and dendritic transcriptomes of single neurons and how both the absolute dendritic abundance and enrichment relative to the soma vary according to gene and function. Finally, we demonstrated that, like glutamatergic neurons, GABAergic neurons translate mRNAs in their dendritic compartments.

The subcellular scRNA-seq method presented here combines LCM to isolate cellular compartments from a single neuron with scRNA-seq to profile the RNAs within it. To decrease the labor-intensive nature of LCM protocols, reduce technical variability, increase sensitivity for low-input samples, and enable the use of UMI, we adopted and modified a library preparation method initially developed for droplet microfluidics (*Macosko et al., 2015*). This allowed us to combine single subcellular compartments into fewer samples and use RNA-sequencing to reconstruct the relation between molecules and subcellular compartments. This technique has several limitations. First, our profiles likely represented only a portion of the mRNAs actually present in dendrites. We estimate that our method detects mRNA species present in as low as four copies in the material collected by LCM, but additional losses likely occur before collection due to RNA degradation and the failure to gather all cut pieces. Accordingly, our dendritic profiles were relatively shallow and may have failed to detect genes present in moderate-to-low copy numbers. This method would thus be inadequate to profile mRNAs within individual dendritic branches. Instead, emerging methods in spatial transcriptomics relying on in-situ hybridization are better equipped for such questions (*Eng et al., 2019*; *Liao et al., 2021*; *Xia et al., 2019*). In contrast to FISH-based transcriptomic techniques, however, our method is unbiased (does not rely on a pre-determined set of transcripts), bypasses secondary structure that may limit accessibility for certain mRNAs in situ, and does not suffer from signal saturation of highly abundant transcripts. Therefore, it may be a good option for transcript discovery where a priori information about the sequence or expression levels of transcripts is lacking (e.g. exploring the usage of alternative 3'UTRs or when working in less annotated cell types or species).

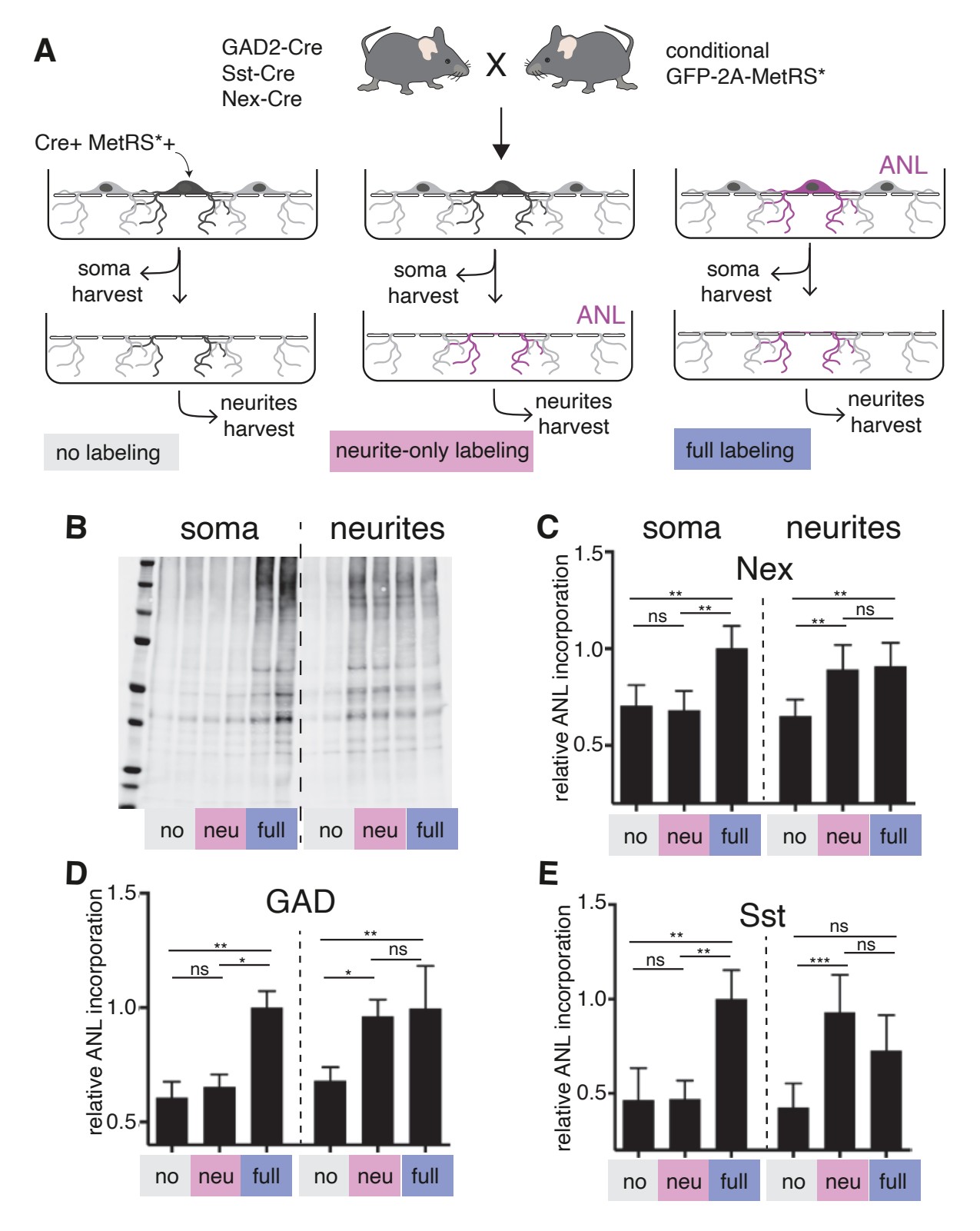

**Figure 6.** Local protein synthesis occurs in the dendrites of GABAergic neurons. (A) Schematic outline of the experiment addressing translational capacity of GABAergic neurites (in GAD-Cre::GFP-2A-MetRS*, Sst-Cre::GFP-2A-MetRS*) and glutamatergic neurites (Nex-Cre::GFP-2A-MetRS*) with cell type-specific ANL incorporation and BONCAT for results shown in B–E. (B) Representative biotin immunoblot for an experiment with Sst-Cre::GFP-2A-MetRS* mouse cortical neurons as outlined in A. (C–E) Quantification of the relative biotin western blot signal representing 15 min ANL incorporation in

*Figure 6 continued on next page*

*Figure 6 continued*

the soma (left bars) or neurite (right bars) fractions harvested under the conditions shown in (**A**) from cortical neuron cultures prepared from the indicated mouse lines: (**C**) GAD-Cre::GFP-2A-MetRS*, (**D**) Sst-Cre::GFP-2A-MetRS*, and (**E**) Nex-Cre::GFP-2A-MetRS* (mean ± SD; GAD: soma no 0.61 ±0.07, soma neu 0.65 ± 0.06, soma full 1.00 ± 0.07, neurites no 0.68 ± 0.06, neurites neu 0.96 ± 0.07, neurites full 1.00 ± 0.19; Sst: soma no 0.47 ± 0.17, soma neu 0.47 ± 0.1, soma full 1.00 ± 0.15, neurites no 0.43 ± 0.13, neurites neu 0.93 ± 0.2, neurites full 0.73 ± 0.19; Nex: soma no 0.71 ± 0.11, soma neu 0.68 ± 0.10, soma full 1.00 ± 0.12, neurites no 0.65 ± 0.09, neurites neu 0.89 ± 0.13, neurites full 0.91 ± 0.12). Equal protein amounts were loaded within soma fractions and within neurite fractions. The biotin immunoreactivity in the neurite fraction of the neurite-only labeling from all investigated mouse lines was significantly higher than in unlabeled controls ([Kruskal–Wallis test with Dunn's multiple comparison's test for GAD, n = 6 (three experiments); GAD soma: soma no vs soma neu p>0.9999, soma no vs soma full p=0.0032, soma neu vs soma full p=0.0304; GAD neurites: neurites no vs neurites neu p=0.0239, neurites no vs neurites full p=0.0073, neurites neu vs neurites full p>0.9999; Sst n = 7 (three experiments); Sst soma: soma no vs soma neu p>0.9999, soma no vs soma full p=0.0023, soma neu vs soma full p=0.0089; Sst neurites: neurites no vs neurites neu p=0.0008, neurites no vs neurites full p=0.0635, neurites neu vs neurites full p=0.5449]; Nex n = 8 [four experiments] Nex soma: soma no vs soma neu p>0.9999, soma no vs soma full p=0.0036, soma neu vs soma full p=0.0017; Nex neurites: neurites no vs neurites neu p=0.0050, neurites no vs neurites full p=0.0080, neurites neu vs neurites full p>0.9999).

While studies of neuronal mRNA localization have generated several lists of localized mRNAs (**Holt et al., 2019**) much less attention has been paid to potential cell-to-cell diversity. To investigate the role cell identity has on the dendritic transcriptome, we determined the cell types present in hippocampal cultures based on the somatic transcriptome. This revealed two types of glutamatergic and three types of GABAergic neurons, although more subtypes likely exist within these categories. We detected some dendritic transcripts that varied according to cell identity. These included some of the genes expressed in a cell type-specific manner, and, surprisingly, genes expressed at similar levels across different cell types (based on somata transcripts). This latter group suggests differences in the machinery regulating dendritic localization according to cell type. Transport of mRNAs to dendrites is known to rely on interactions between RNA elements, particularly in 3'UTRs, and RNA binding proteins, thus enabling their movement along microtubules (**Darnell, 2013**; **Glock et al., 2017**). Cell type-driven variations in mRNA localization could be explained by the cell type-specific expression of proteins capable of modifying RNA elements, like RNA-editing enzymes or epitranscriptome regulators (**Behm and Öhman, 2016**; **Merkurjev et al., 2018**), or of RNA binding proteins that may activate dendritic transport of mRNAs containing its targeted elements (**Martin and Ephrussi, 2009**). Significantly more cell type-specific differences were observed between somata than between dendritic arbors, based on both unsupervised clustering and differential expression analyses. This is to be expected since the soma harbors the dendritic transcriptome in addition to transcripts that maintain cell identity (e.g. transcription factors) and axonal specialization (e.g. neurotransmitter(s) machinery). While it is likely that our method lacks sensitivity, it is also conceivable that less transcriptomic heterogeneity occurs between the dendrites of different neurons for biological reasons. Dendrites, as the recipients and integrators of excitatory and inhibitory synaptic inputs, may be more generalized than other cellular compartments. Some cell type-specific dendritic proteins may be synthesized in somata. Additionally, given the complex signal integration and plasticity that occurs within dendrites, it is plausible that cell states have bigger influences than cell types in the local mRNA repertoire.

We also investigated the patterns of mRNA distribution between the somatic and dendritic compartments of single neurons. We observed that expression levels established in the soma positively influence the dendritic presence and abundance of some, but not all, mRNAs and that the magnitude of this effect varies significantly according to the transcript. By comparing the expression between the dendrites and soma of the same neuron we identified hundreds of genes relatively enriched and de-enriched in dendrites, which occurred at a wide range of absolute abundances in dendrites. Our results are consistent with a model in which mRNAs compete for access to the transport machinery creating an advantage for the more abundant mRNA species, like *Calm1* or *Atp5f1b*. To compensate, less abundant mRNA species like *Ccni* and some ribosomal proteins might increase their affinity for the transport machinery based on the type or number of elements in their sequence. Interestingly, we also observed some mRNAs like *Gria2* and *Psmb2* that were significantly de-enriched in dendrites despite their moderate-to-high somatic abundances. Although it is possible that these mRNAs simply lack the elements that allow dendritic localization, it is also possible that mechanisms exist to prevent dendritic entry or to ensure somatic confinement.

Using GO analyses, we identified multiple cellular functions highly represented in the somatic and dendritic transcriptomes. Mitochondrial functions were the most represented among highly abundant somatic and dendritic genes, consistent with the high energetic demands of neuronal compartments. Indeed, local production of mitochondrial proteins was recently implicated in the function of local mitochondria (*Kuzniewska et al., 2020*), which supply energy to spatially stable compartments containing multiple synapses (*Rangaraju et al., 2019*). Intriguingly, we also observed contrasting patterns between the protein synthesis and the protein degradation machinery. Ribosomal protein (RP) mRNAs were detected across a wide range of abundances in both somatic and dendritic compartments, as observed previously (*Cajigas et al., 2012*; *Gumy et al., 2011*; *Poon et al., 2006*). Despite this high variance in absolute expression most RP mRNAs exhibited dendritic enrichments relative to somatic expression of similar magnitudes. In contrast, most proteasome genes were uniformly de-enriched in dendrites despite having diverse somatic expression values. A similar opposing trend in mRNA localization between ribosomal and proteasome genes was observed in the growth cone of cortical projection neurons (*Poulopoulos et al., 2019*).

Finally, the dendritic mRNAs and the intracellular localization dynamics described above were observed in both GABAergic and glutamatergic neurons, suggesting both can locally synthesize proteins. Indeed, we observed that GABAergic neurons localize thousands of mRNAs to their dendrites and showed directly that they also perform local translation, suggesting this is a general property of neuronal dendrites. Consistently, a recent study described the local synthesis of protein in an inhibitory axon terminal during plasticity (*Younts et al., 2016*). Moving forward, it will be important to investigate how plasticity and changes in cell state regulate the dendritic transcriptome of individual neurons, ideally in the context of neuronal circuits. For instance, glutamatergic and GABAergic neurons within a circuit establish an excitatory–inhibitory balance that is crucial for proper brain functions (*Nelson and Valakh, 2015*). Single cell resolution of the local transcriptome will bring a better understanding of the intracellular mechanisms regulating its properties, as well as better comprehension of the dendritic and axonal compartments participating in diverse neuronal circuits.

# Materials and methods

## Key resources table

| Reagent type (species) or resource | Designation | Source or reference | Identifiers | Additional information |
|---|---|---|---|---|
| Genetic reagent (*Rattus norvegicus*) | Primary neurons | Other | | Animals obtained from our animal house |
| Genetic reagent *Mus musculus* | Primary neurons | Other | | Animals obtained from our animal house |
| Genetic reagent (*M. musculus*) | Sst-IRES-Cre | Jackson | JAX013044 | |
| Genetic reagent (*M. musculus*) | Gad2-IRES-Cre | Jackson | JAX010802 | |
| Genetic reagent (*M. musculus*) | Nex-Cre | Obtained from the laboratory of Dr. Klaus-Armin Nave | | *Goebbels et al., 2006* PMID:17146780 |
| Genetic reagent (*M. musculus*) | STOPflox R26-GFP-2A-MetRS*(L274G) | Developed in house | JAX028071 | |
| Antibody | Rabbit polyclonal anti-biotin | SIGMA | 31852 | (1:1000) IB |
| Antibody | Guinea pig polyclonal anti-MAP2 | Synaptic Systems | 188004 | (1:1000) IF |
| Antibody | Goat polyclonal anti-rabbit IRDye680 | Licor | Goat anti-rabbit IRDye680 | (1:5000) IB |
| Antibody | Goat polyclonal anti-guinea pig Dylight405 | Jackson ImmunoResearch | Goat anti-guinea pig Dylight405 | (1:500-1:1000) IF |

*Continued on next page*

*Continued*

| Reagent type (species) or resource | Designation | Source or reference | Identifiers | Additional information |
|---|---|---|---|---|
| Antibody | Goat polyclonal anti-guinea pig-Alexa488 | Thermo-fisher | Goat anti-guinea pig-Alexa488 | (1:1000) IF |
| Commercial assay or kit | Nextera XT DNA Library prep kit | Illumina | FC-131–1096 | |
| Commercial assay or kit | NextSeq 500/550 High Output Kit v2.5 (75 Cycles) | Illumina | 20024906 | |
| Commercial assay or kit | ViewRNA ISH Cell Assay Kit | Thermo-fisher | QVC0001 | |
| Software, algorithm | Cufflinks 2.2.1 | Cufflinks 2.2.1 | | Annotation of Rat 3'UTRs |
| Software, algorithm | Drop-seq tools (v2.3.0) | Drop-seq tools (v2.3.0) | | Sequencing data processing |
| Software, algorithm | Fastp | Fastp (v0.20.1) | | Read quality control |
| Software, algorithm | STAR 2.5.4b | STAR 2.5.4b | | Genome alignments |
| Software, algorithm | Seurat 3.1.5 | Seurat 3.1.5 | | Data analyses |
| Software, algorithm | GSEA_4.0.3 | GSEA_4.0.3 | | Pathway enrichment analyses |
| Software, algorithm | Fiji ImageJ Version 2.0.0-rc-68/1.52 n | Fiji ImageJ Version 2.0.0-rc-68/1.52 n | | FISH data analyses |

## Animals

All animals were housed under conditions approved by the local governmental authorities in standard cages under a 12 hr:12 hr light–dark cycle with standard lab chow and water ad libitum. Female Crl:CD(SD) rats with timed matings were purchased from Charles River Laboratories (Sulzfeld, Germany) (specific pathogen free [spf] colonies). Newborn rat pups of either sex were used for preparation of hippocampal neuron cultures. Transgenic Cre-driver mouse lines were purchased from the Jackson laboratory (Sst-IRES-Cre [JAX 013044, *Taniguchi et al., 2011*], GAD2-IRES-Cre [JAX 010802, *Taniguchi et al., 2011*]) or kindly provided by K.A. Nave (Nex-Cre [*Goebbels et al., 2006*]), the conditional STOPflox R26-GFP-2A-MetRS*(L274G) mouse line was developed in house and is available from Jackson Labs (*Alvarez-Castelao et al., 2017*, JAX 028071), and all lines were kept under spf conditions. Homozygous females of the Cre-driver lines were crossed to STOPflox R26-GFP-2A-MetRS*(L274G) males and neuron cultures were prepared from cortices and hippocampi of newborn animals without determination of their sex. The procedures involving animal treatment and care were conducted in conformity with the institutional guidelines that are in compliance with the national and international laws and policies (DIRECTIVE2010/63/EU; German animal welfare law, FELASA guidelines) and approved by and reported to the local governmental supervising authorities (Regierungspräsidium Darmstadt). The animals were euthanized according to annex 2 of §2 Abs. 2 Tierschutz-Versuchstier-Verordnung.

## Primary neuronal cultures

Neuronal cultures were prepared and maintained essentially as described (*Aakalu et al., 2001*). Briefly, the hippocampus of P1 rat pups or cortex and hippocampus of P0/P1 mice of either sex was dissected out, dissociated with papain (Sigma), and plated on poly-D-lysine coated coverslips of MatTek glass bottom dishes (P35G-1.5–14 C) at a density of 20,000 and 30,000 cells/coverslip for LCM and FISH experiments, respectively. For experiments on transwell cell culture inserts mouse cortical neurons were seeded onto 3 µm pore size 75 mm diameter polycarbonate membranes (15 Mio) (Costar) or 6-well format PET inserts (1.2 Mio) (Corning) coated with poly-D lysine. All neurons were maintained at 37°C and 5% $CO_2$ in glia- and cortex-conditioned Neurobasal-A (Life Technologies) supplemented with B27 and GlutaMax. Neurons for LCM and FISH experiments were used 2 weeks and 3 weeks after plating, respectively.

## Laser capture microdissection

All LCM experiments were conducted with Zeiss's PALM MicroBeam Microscope. Cells were fixed for 5 min using 70% ethanol at −20°C and stored at −80°C until the day of experiment. Single neurons were selected, imaged, and dissected under a 40× objective. Dissection was performed by Laser Pressure Catapulting, with energy of 40, focus of 70% and 15% of speed. The soma of the neuron was always dissected first, followed by dissection of all its accessible dendrites, capturing all accessible segments irrespective of bifurcations (*Figure 1—figure supplement 1C*). On average, collected neurons had five accessible primary dendrites. Cells were collected from 12 different plates from two separate culture preparations.

## Library preparation and sequencing

Material catapulted by LCM was collected in 3 µL of lysis buffer containing: 1× SingleShot lysis buffer (Bio-rad), 45 mAU/mL Qiagen Protease (Qiagen), 10 mM dNTPs (Thermo-fisher), 0.1 µM of 1 out of 18 different RT custom primers containing an Index and UMI (*Supplementary file 7*), and ERCC RNA standards (Thermo-fisher) diluted 1:5 × 10$^6$. Samples were then incubated at 50°C for 10 min for protein digestion and 75°C for 10 min to inactivate the protease. Reverse transcription (RT) and template switch buffer containing 10 U/µL of SuperScript IV and its buffer at 1× (Thermo-fisher), 40 U/µL RNAse Inhibitor (Takara Bio), 1 µM Template Switch Oligo (*Supplementary file 7*), 6 mM MgCl$_2$ (Thermo-fisher), 5 mM DTT (Bio-rad), and 1M betaine (Sigma) was then added to the digestion reaction for a final volume of 6.4 µL and incubated at 55°C for 10 min of RT, followed by a 10 min incubation to inactivate the RT enzyme. Next, samples underwent a pre-amplification PCR reaction containing 1× KAPA Hifi Hot-Star Mix (Roche) and 0.1 µM ISPCR (*Supplementary file 7*) for a final volume of 14 µL. Then, samples with different indexes but of the same source (somata, dendrites, or empty cuts) were pooled together in groups of eight and purified using AMPure XP beads (Beckman Coulter). Libraries were further indexed, prepared, and pooled a final time using the Nextera XT DNA library prep kit (Illumina), according to manufacturer's instructions. Library amplification PCR was performed with a custom P5 primer (*Supplementary file 6*). Finally, paired-end sequencing was conducted on the NextSeq 550 (Illumina) according to the following specifications: Read 1 = 18 bp (Index and UMI), Read 2 = 58 bp (gene), and Read 1 Index = 8 bp (i7 index). Read 1 was generated using a custom primer (*Supplementary file 7*). Samples were sequence in four separate runs.

## Transcript annotations

The rat transcriptome is not as well annotated as the mouse or human transcriptome especially in its untranslated regions. This is particularly limiting when using 3'end sequencing (like the subcellular scRNA-seq method used here) to analyze mRNAs localized to neurites since they tend to have longer 3'UTRs that are often unannotated (*Tushev et al., 2018*). Thus, to improve our detection of mRNAs we used a recent full-length reconstruction of the hippocampal rat transcriptome using long read sequencing (*Wang et al., 2019*). We then created a Merged Transcriptome, by combining this full-length annotation with non-overlapping Ref-seq and Ensemble annotations. Finally, we performed bulk RNA-sequencing on the same plates used for subcellular scRNA-seq and used Cufflinks 2.2.1 (*Trapnell et al., 2012*) to identify additional alternative 3'UTRs. Cufflinks assembly was conducted using the Merged Transcriptome as a reference guide and included only additions to genes present in the reference, that is, novel genes were discarded.

## Sequence quality filters, read mapping, and generation of digital expression data

Sequencing data was processed according to the Drop-seq core computation protocol (*Macosko et al., 2015*). Briefly, using Drop-seq tools (v2.3.0) and Fastp (*Chen et al., 2018*) reads were removed if they contained low quality indexes or UMI, or low quality, low complexity, or too short (<30 bp) gene sequences. Additionally, the ends of the reads were trimmed when containing adapter sequences or stretches of >6 nt of the same base (e.g. Poly A). Reads were aligned to the Rat genome using STAR 2.5.4b. Reads with multiple alignments or excessively soft-clipped (>28 bp) were removed. Reads were then assigned to genes, based on their overlapped with the transcript annotations described above. Finally, UMIs of the same gene and same sample (same index) within edit distance of 1 were merged, and counted to determine the expression of each genes within each

sample. Mitochondrial encoded mRNAs were excluded from all expression analyses as they are locally synthesized and thus beyond the scope of this study. Tables containing mRNA counts per sample were imported into R and analyzed using Seurat 3.1.5.

## Normalization of UMI counts

All analyses, unless otherwise specified, were performed on 'Normalized Expression' which was calculated using Seurat's SCTransform function (*Hafemeister and Satija, 2019*). Briefly, SCTransform uses the residuals of a regularized negative binomial regression to correct for technical variation on UMI counts. In UMAP and Heatmap plots the natural logarithm of normalized expression is shown.

## Dimensionality reduction and cluster identification

Dimensionality reduction consisted of determining the principal components (PC) of the data using Seurat's RunPCA function and then using the top 40 PCs for UMAP embedding using Seurat's RunU-MAP function. To identify clusters, we used the top 23 PCs to construct a k-nearest neighbor graph of each cell, which was then used to construct a shared nearest neighbor graph by calculating the overlap between k-nearest neighbors, using Seurat's FindNeighbors function. Modularity was optimized using the Louvain method (*Blondel et al., 2008*) in Seurat's FindClusters function.

## Differential expression

To determine gene markers of cell types identified in the somatic data set we used the Seurat FindAllMarkers function to conduct a logistic regression test (*Ntranos et al., 2019*) on genes expressed in at least 25% of samples and with a log fold-change above 0.25. A cutoff of adjusted p-value <0.05 was used to determine the significance. Differential expression analyses between somata or dendrites of different cell types were conducted by the above-mentioned logistic regression test using genes detected in at least 10% of the samples and with a log fold-change above 0.5, using Seurat's FindMarkers function (*Figure 3A*). A cutoff of adjusted p-value <0.05 was used to determine significance in somata samples. In dendrites, however, this same cutoff reported almost no significant genes in all of the comparisons. To determine whether adjusted p-value of 0.05 was overly conservative when working with a shallower transcriptome and fewer samples (as is the case for dendrites) we performed three iterations of differential expression testing on a number of somatic samples equal to dendritic samples, and which were downsampled to equivalent molecular counts (data not shown). In these conditions, an adjusted p-value cutoff of 0.05 missed most somatic differential expression between cell types. To calculate a more accurate significance cutoff we used a bootstrapping mock test approach. Briefly, dendrites belonging to each of the two groups being compared were randomly shuffled to create two mock groups while preserving the same number of samples that were present in the real groups. Differentially expressed genes in the mock groups were then identified using the same test and parameters used in the real comparison. For each cell type comparison, we tested 1000 mock permutations and create an average distribution of p-values (*Figure 2—figure supplement 1C*). This distribution was then used to identify the p-value in which a 5% false discovery rate was obtained (*Figure 2—figure supplement 1D*). This p-value was then used as the cutoff in the real data set. Paired-differential expression analyses between soma and dendrites of the same neuron were conducted with Seurat's FindMarkers function for genes expressed in at least 25% of samples using a Poisson generalized linear model (*Stuart et al., 2019*) with cell of origin as a latent variable. A cutoff of adjusted p-value <0.05 was used to determine significance.

## Integration with hippocampal tissue scRNA-seq

Tables containing gene counts per cell were obtained from *Middleton et al., 2019*, *Zeisel et al., 2015*, and *Harris et al., 2018* supplementary information, and run through data normalization. The Zeisel et al. and Harris et al. data sets also underwent dimensionality reduction, cluster identification, and differential expression as previously discussed. This allowed us to match the clusters identified in our analysis with those reported by the respective authors. Since all these data sets are derived from *Mus musculus*, for integration analyses we converted the common gene names of *Rattus norvegicus* to that of their mouse ortholog, if available, using Ensembl annotations. Integration of data sets with our somata samples was performed as previously described (*Stuart et al., 2019*), using canonical correlation analyses to find shared subpopulations across data sets.

## Gene ontology analyses

Gene lists pre-ranked by somatic or dendritic 'representation', which equal the product of detection frequency $\times$ expression levels for each compartment, were used in a gene set enrichment analysis (GSEA, [Subramanian et al., 2005]), against GO cellular components and SynGO terms (Koopmans et al., 2019). GSEA was performed using a weighted scoring scheme, and only terms containing less than 500 genes and with at least 15 genes present in our data were evaluated.

## Immunocytochemistry

For immunocytochemistry, cells were permeabilized for 15 min with 0.5% Triton in blocking buffer (BB) (PBS with 4% goat serum), blocked in BB for 1 hr and incubated with primary antibodies in BB for 1 hr at room temperature. After washing, secondary antibodies in BB were applied for 30 min followed, when necessary, by a 3 min incubation with 1 µg/µL DAPI in PBS to stain nuclei. Cells were washed in PBS and mounted with Aquapolymount (Polysciences).

## High sensitivity RNA-FISH

In situ hybridization was performed using the ViewRNA ISH Cell Assay Kit (Thermo-fisher) according to the manufacturer's protocol with the modifications described previously (Cajigas et al., 2012). Probe sets targeting the respective mRNAs were purchased from Thermo-fisher. In brief, rat hippocampal or mouse cortical/hippocampal neuron cultures grown on MatTek glass bottom dishes were fixed for 20 min with PBS containing 4% sucrose and 4% PFA, pH 7.4, washed and permeabilized for 5 min with the provided detergent solution. Gene-specific type 1 and type 6 probe sets were applied in 1:100 dilution for 3 hr at 40°C. Signal amplification steps with PreAmp/Amp and Label Probe reagents coupled with 550dye (and 650dye or 488dye in dual color FISH) were all performed for 1 hr at 40°C with branched DNA reagents diluted in the provided solutions 1:100. Staining for markers was performed after the FISH protocol as described in the immunocytochemistry section starting after the permeabilization step. For experiments in combination with FUNCAT the FISH protocol was performed after the metabolic labeling step and the click reaction applied after finishing the FISH protocol.

## Metabolic labeling and FUNCAT

Mouse neurons expressing the MetRS*(L274G) mutant under control of GAD2-IRES-Cre, Sst-IRES-Cre, or Nex-Cre, respectively, were maintained on MatTek glass bottom dishes at 37°C and 5% $CO_2$ in glia- and neuron-conditioned Neurobasal-A with B27 and GlutaMax supplements. For metabolic labeling with ANL the cells were washed for 5 min with Neurobasal A with the above supplements but lacking methionine, incubated for 60 min with 4 mM ANL in Neurobasal A$^{-Met}$ with supplements, and finally washed with their own original methionine-containing medium for 5 min at 37°C 5% $CO_2$ and humidified atmosphere and fixed after another brief washing step as described in the RNA-FISH section. After performing the FISH protocol, the cells were blocked for 30 min with 4% goat serum in PBS pH 7.4, equilibrated to PBS pH 7.8, and subjected for 2 hr to a copper mediated click reaction in PBS pH 7.8 at RT (with the following reagents: 200 µM Triazole, 500 µM TCEP, 2 µM Alexa647 alkyne, 200 µM CuSO4) before proceeding with immunocytochemistry for marker proteins.

## Metabolic labeling of neurites and BONCAT

Mouse neurons on transwell inserts were maintained at 37°C and 5% $CO_2$ in glia- and cortex-conditioned Neurobasal-A supplemented with B27 and GlutaMax until 10 days in vitro. For the experiment the inserts were carefully rinsed two times in Neurobasal A lacking methionine (Neurobasal A$^{-Met}$, Life Technologies) and incubated with 8 mM ANL in Neurobasal A$^{-Met}$ with B27 and Glutamax supplements for 15 min at 37°C with 5% $CO_2$ in a humidified atmosphere with or without prior removal of the soma layer by scraping (above the membrane). Negative control samples were treated in the same way but without ANL added to the media. Soma and neurite layers were harvested separately by two times scraping in PBS, transferred to a microtube followed by a 30 s centrifugation in a benchtop minifuge (neoLab, Heidelberg, Germany). When 6-well transwell membranes were used, material from three wells was combined for one replicate sample. After spindown, the supernatant was removed, the pellet immediately frozen on dry ice, and stored at −80°C until lysis.

The samples were lysed in a minimal volume of PBS with 1% (w/v) TritonX-100, 0.4% (w/v) SDS, protease inhibitors w/o EDTA (Calbiochem, 1:750), and benzonase (Sigma, 1:1000), heated to 75°C for 5 min and centrifuged. Supernatants were analyzed and adjusted for equal protein content within the soma samples and within the neurite samples of the experiment. BONCAT was performed as described previously (*Dieterich et al., 2007*). In brief, 20 μL of the sample was subjected to a click reaction with 300 μM Triazol (Sigma, ref 678937), 50 μM biotin-alkyne tag (Thermo-fisher), and 83 μg/mL CuBr at 4°C overnight in the dark in PBS pH 7.8. SDS-PAGE was performed and proteins were subsequently blotted onto PVDF membranes. Biotinylated proteins were detected by Immunoblot with a biotin antibody and IRDye680-coupled secondary antibody and scanned with a fluorescence-based detection system (Odyssey, LICOR, Bad Homburg, Germany).

### Western blot analysis

Corresponding regions-of-interest from the same western blot were marked in each lane in Fiji ImageJ Version 2.0.0-rc-68/1.52 n (NIH); mean gray values were determined for each sample and converted to fraction of the mean value of the full labeling controls in somata for each blot. Biological replicates were from ≥3 independent experiments for each of the three mouse lines. Statistical significance of the differences within soma fractions from the different conditions and the neurite fractions from the different conditions were determined for every line by ANOVA (Kruskal–Wallis test corrected for multiple comparisons with Dunn's multiple comparison's test) in GraphPad Prism 6.

### ANL synthesis

ANL was synthesized from Boc-L-Lys-OH (Iris Biotech, Marktredwitz, Germany), Triflic anhydride trifluoromethanesulfonic acid anhydride, Merck/Sigma, and sodium azide in analogy to the synthesis of AHA from Boc-Dab described by *Link et al., 2007*.

### Antibodies

The following antibodies were used for immunofluorescence (IF) and immunoblotting (IB) at the indicated dilutions: rabbit anti-biotin (IB 1:1000, Sigma), guinea pig anti-MAP2 (IF 1:1000, Synaptic Systems), anti-rabbit IRDye680 (IB 1:5000, Licor), goat anti-guinea pig Dylight405 (IF 1:500-1:1000, Jackson ImmunoResearch), and goat anti-guinea pig-Alexa488 (IF 1:1000, Thermo-fisher).

### Confocal microscopy

Images were acquired with a LSM780 confocal microscope (Zeiss) using a 40×/1.4-NA oil objective (Plan Apochromat 40×/1.4 oil DIC M27) and appropriate laser lines set in single tracks and pixel resolution of at least 2048 × 2048. All lasers were used at 2% power or less. Images were acquired in 12-bit or 16-bit mode as *z* stacks with optical slice thickness set to optimal and the detector gain in each channel adjusted to cover the full dynamic range but to avoid saturated pixels. Maximum intensity projections were created in Zen10 software or in Fiji/ImageJ. For visualization, linear adjustments of the single channels for brightness over the whole image were performed. Conditions were kept constant within an experiment.

### Soma FISH analysis

From maximum intensity projections of FISH confocal images somata of all neurons in the field of view were outlined in Fiji, mean gray values were measured in all ROIs representing somata and for each cell mean gray values of cell type marker FISH signal and mRNA of interest FISH signal were plotted in GraphPad Prism 6.

### Dendrite FISH pattern visualization with raster plots

From maximum intensity projections of FISH confocal images dendrites from all neurons in a field of view that could be followed for a minimum length of around 50 μm were traced in Image J/Fiji along the MAP2 label with the segmented line tool starting at the soma until the most distal point where the dendrite was identified without crossing other cells containing signal. Dendrites were straightened, channels split, and converted in profiles of mean intensity values. For each dendrite, signal puncta were detected as peaks in the profile and converted into bar signals at the corresponding

location in a representation of the dendrite with a custom MatLab script. All analyzed dendrite/signal bar representations were stacked in a soma (left) to distal (right) direction to visualize variability and distribution pattern with the end of the tracked dendrite marked in the plot. In *Figure 3* dendrites were sorted prior to plotting based on a threshold level in a second channel stained for the cell type marker mRNAs Gad1/2. For FISH experiments in *Figure 4F*, neuron identity was tracked by cell type-specific ANL incorporation into mouse cortical neurons (from mouse lines described in *Figure 6*) and FUNCAT. To ensure coverage of excitatory and inhibitory neuron dendrites in the raster plots, cells with and without FUNCAT signal were imaged and dendrites straightened.

## Analyses of cell morphology

We used ImageJ and PolygonRoi PlugIn to draw regions of interest. On every image a mask with the cell soma, cell nucleus, and each primary dendrite thickness was drawn. The mask was used to define the following morphological features: soma and nucleus area, primary dendrite thickness, median intensity in soma and nucleus. To avoid bias in intensity levels we defined a relative intensity measure by dividing soma vs nucleus median intensities. Morphological features were compared using non-parametric ANOVA test (Kruskal–Wallis test). The resulting ANOVA table was used in a multi-comparison test (Dunn's test) to assess each cell type class differences.

## Acknowledgements

We thank Ina Bartnik, Dirk Vogel, Bernadette Tune, Matt White, and Michelle Gottlieb for assistance with experiments and analyses, Or Shahar and Stephan Junek for help with the LCM microscope, and Belquis Nassim-Assir for help with BONCAT experiments. BAC is funded by Comunidad de Madrid (Atracción de Talento-2019T1/BMD-14057) and Ministerio de Ciencia e Innovación (Ramón y Cajal- RYC2018-024435-I). EMS is funded by the Max Planck Society, an Advanced Investigator award from the European Research Council (grant 743216), DFG CRC 1080: Molecular and Cellular Mechanisms of Neural Homeostasis, and DFG CRC 902: Molecular Principles of RNA-based Regulation.

## Additional information

### Funding

| Funder | Grant reference number | Author |
| --- | --- | --- |
| H2020 European Research Council | | Erin M Schuman |
| Comunidad de Madrid | Atraccion de Talento-2019T1/BMD-14057 | Beatriz Alvarez-Castelao |
| Ministerio de Ciencia e Innovación | Ramón y Cajal- RYC2018-024435-I | Beatriz Alvarez-Castelao |
| European Research Council | grant 743216 | Erin M Schuman |
| Deutsche Forschungsgemeinschaft | DFG CRC 1080 | Erin M Schuman |
| Deutsche Forschungsgemeinschaft | DFG CRC 902 | Erin M Schuman |

The funders had no role in study design, data collection and interpretation, or the decision to submit the work for publication.

### Author contributions

Julio D Perez, Conceptualization, Data curation, Software, Formal analysis, Investigation, Visualization, Methodology, Writing - original draft; Susanne tom Dieck, Investigation, Visualization, Methodology; Beatriz Alvarez-Castelao, Data curation, Formal analysis, Investigation; Georgi Tushev, Data curation, Software, Formal analysis; Ivy CW Chan, Data curation, Investigation; Erin M Schuman,

Conceptualization, Supervision, Funding acquisition, Project administration, Writing - review and editing

## Author ORCIDs
Julio D Perez ![ORCID] https://orcid.org/0000-0002-8769-9306
Susanne tom Dieck ![ORCID] https://orcid.org/0000-0002-5884-8640
Beatriz Alvarez-Castelao ![ORCID] http://orcid.org/0000-0001-7505-1855
Erin M Schuman ![ORCID] https://orcid.org/0000-0002-7053-1005

## Ethics
Animal experimentation: The procedures involving animal treatment and care were conducted in conformity with the institutional guidelines that are in compliance with the national and international laws and policies (DIRECTIVE2010/63/EU; German animal welfare law, FELASA guidelines) and approved by and reported to the local governmental supervising authorities (Regierungspräsidium Darmstadt). The animals were euthanized according to annex 2 of §2 Abs. 2 Tierschutz-Versuchstier-Verordnung.

## Decision letter and Author response
Decision letter https://doi.org/10.7554/eLife.63092.sa1
Author response https://doi.org/10.7554/eLife.63092.sa2

# Additional files

## Supplementary files
• Supplementary file 1. Sample metadata. Related to *Figures 1–4*. Contains information on the experimental design, quality metrics, and experimental variables for each subcellular single cell RNA-seq (scRNA-seq) sample.

• Supplementary file 2. Count data. Related to *Figures 1–4*. Contains unique molecular identifier (UMI) counts in a gene (rows) by samples (columns) table.

• Supplementary file 3. Somatic and dendritic transcriptomes of GABAergic and glutamatergic neurons. Related to *Figure 2*. Contains average normalized unique molecular identifier (UMI) counts of genes detected in the dendrites (Tab 1) and somata (Tab 2) of GABAergic and glutamatergic neurons.

• Supplementary file 4. Somata and dendrites cell type-specific differential expression results. Related to *Figure 3*. Contains results of differential expression test for: (Tab 1) GABAergic vs glutamatergic somata, (Tab 2) GABAergic vs glutamatergic dendrites, (Tab 3) GABAergic 1 vs GABAergic 2 and 3 somata, (Tab 4) GABAergic 1 vs GABAergic 2 and 3 dendrites, (Tab 5) GABAergic 2 vs GABAergic 3 somata, (Tab 6) GABAergic 1 vs GABAergic 2 and 3 dendrites.

• Supplementary file 5. Somata vs dendrites differential expression results. Related to *Figure 4*. Contains results of somata vs dendrites differential expression test.

• Supplementary file 6. Gene ontology analysis. Related to *Figure 5*. Contains results of GSEA ranked test for somata (Tab 1) and dendrites (Tab 2).

• Supplementary file 7. Sequences of primers used in this study. Related to methods.

• Transparent reporting form

## Data availability
Sequencing data have been deposited in GEO under accession code GSE157204.

The following dataset was generated:

| Author(s) | Year | Dataset title | Dataset URL | Database and Identifier |
|---|---|---|---|---|
| Perez JD, Schuman EM | 2021 | Single Cell Sequencing: mRNA Diversity of Individual Glutamatergic and GABAergic Hippocampal Neurons and their Dendrites | https://www.ncbi.nlm.nih.gov/geo/query/acc.cgi?acc=GSE157204 | NCBI Gene Expression Omnibus, GSE157204 |

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
