## [Decision Letter]

**Acceptance summary:**

This manuscript is an excellent resource for comparing gene expression in dendrites versus soma in hippocampal neurons. It contains important information regarding dendritic RNA localization in the two major classes of neurons.

**Decision letter after peer review:**

Thank you for submitting your article "Subcellular sequencing of single neurons reveals the dendritic transcriptome of GABAergic interneurons" for consideration by *eLife*. Your article has been reviewed by three peer reviewers, including Genevieve Konopka as the Reviewing Editor and Reviewer #1, and the evaluation has been overseen by Laura Colgin as the Senior Editor. The following individual involved in review of your submission has agreed to reveal their identity: Eran A Mukamel (Reviewer #3).

The reviewers have discussed the reviews with one another and the Reviewing Editor has drafted this decision to help you prepare a revised submission.

Summary:

All three reviewers were very position about this Tools and Resources article and feel that the dataset and approach will be very valuable for the community. This manuscript details the transcriptome of hippocampal neurons at single cell resolution, comparing transcripts in the soma with transcripts in the dendrites. Comparisons of glutamatergic and GABAergic neurons are also carried out.

Essential revisions:

Each reviewer had distinct suggestions for improving the manuscript including additional analyses of the datasets. Please refer to the individual reviews below and address as many of these comments as possible. If a particular comment cannot be addressed, please state the reason.

Reviewer #1:

This manuscript from Perez and colleagues is a great resource for comparing dendritic versus somatic transcripts in hippocampal neurons at single cell resolution. The authors do an excellent job of describing their experimental details, interpretations of the data, associated caveats, and carry out reasonable confirmations of genomic data using smFISH as well as data supporting local protein synthesis in the dendrites of GABAergic neurons. I was excited to see comparisons of dendritic vs. somatic transcripts within individual cells as well as comparisons of dendritic transcriptomes between cell types. The relative lack of differentiation of the dendritic transcriptome between cell types was somewhat surprising and even though careful downsampling and other means were carried out to account for lower depth in these transcriptomes and the authors discuss potential reasons why this may be, I do wonder if future studies with greater depth will confirm these results and/or if such results might be the case in other brain regions. It might be worthwhile for the authors to compare their data to a study published last year that carried out somatic vs. dendritic profiling in hippocampal neurons: Middleton, Eberwine and Kim, 2019.

Although that study did not differentiate between neuronal cell types, presumably one could examine this using the raw data. It should probably be cited in this manuscript as well.

While I am enthusiastic about the manuscript and the data it will provide to the field, there are some issues that could be addressed to improve the manuscript.

1) In the very first figures the authors should describe how many dendrites are from which category of neurons or maybe highlight them with a different color in the UMAP plot (Figure 1C).

2) A potential deep comparison of the dendritic transcriptome of different cell types seemed exciting to me but was only superficially examined. The authors do mention the DEG list between the different groups, but only name a few genes and then validate them. It would be great to discuss the functional consequence of these differences in how the local dendritic transcriptome difference determines the functioning of these cells during signal transmission. Also, when comparing and integrating the data with published tissue scRNA-seq, why not include the dendritic data too? Such comparisons could be important for others in the field to "deconvolute" their scRNA-seq data to look for dendritic-enriched transcripts.

Reviewer #2:

This paper is excellent with a wealth of data. Some points to consider:

1) In the first Results section they look at hippocampal cultures. The control is empty cuts. Could they perform an axon cut as a control?

2) They give the number of transcripts in the dendritic and somatic compartments. Although discussed in the Introduction and later in the manuscript, did they mention the total RNA extracted in each compartment and look for a selective loss of low abundance dendritic mRNAs normalized to total RNA in the beginning of the paper when making the simple comparisons to transcript numbers in the two compartments?

a) It seems one way they addressed the above issue was by down sampling which is certainly reasonable. But could they determine whether dendritic mRNAs are selectively reduced in the mRNAs from the low abundance tail of somatic mRNAs? This analysis performed at some threshold would offer a simple and quick high level look.

3) The work superbly addresses quantitative comparisons-for example in the Poisson generalized linear model, but some greater to attention to functional differences would enhance the paper. This topic is not ignored-they have a section on "function associations" but that section is very descriptive and not completely as satisfying as one would expect for such profound differences and such an interesting problem. In this vein, their data would be very suitable for a bipartite community detection algorithm. It seems some of the points they want to make concerning dendritic and somatic mRNAs might become more revealing by identifying bipartite modules.

4) They should address parvalbumin cells in some detail.

Reviewer #3:

The paper by Perez and colleagues uses laser capture microdissection of dendrites and somata of cultured rat hippocampal neurons followed by single cell RNA-seq to assess the localization of mRNA transcripts in the somatic and dendritic compartments. This study addresses an important gap in our understanding of the cell type-specific regulation of dendritic RNA localization. Capturing mRNA from dendrites of single cells is challenging, and the dataset and analysis they present convincingly demonstrate that both glutamatergic and GABAergic cell types localize specific mRNA species to the dendrites in support of dendritic protein synthesis.

Overall I found the paper to be well organized and clearly presented. Although the number of cells/dendrites and the number of transcripts per cell/dendrite are modest compared with scRNA-seq studies, due to the challenge of manual LCM, the data quality appears to be sufficient to clearly separate at least ~2 glutamatergic and 3 GABAergic cell types.

1) The authors compare rat cultured neurons, derived from P0 animals, with large-scale scRNA-seq datasets from adult mouse primary hippocampus samples. What are the caveats from the species difference and also the difference between cultured and primary neurons? This should be at least mentioned.

2) – Data about gene expression are presented without any units (e.g. Figure 2A, B – y-axis; Figure 3A, B, Figure 4A-C). Are these values TPM, FPKM, CPM, or something else? In the axes with log units, is it showing log2 or log10?

3) – In the differential expression analysis, the authors explain that they used an adjusted p-value (i.e. FDR control) for somatic cell type differences, but they chose an uncorrected p-value cutoff (p<.02) for the dendrites. The reasoning they provide is that the FDR-control is "overly punitive" (meaning conservative) and would cause a high rate of false negatives, which they demonstrate by downsampling the somatic datasets. This is not a statistically sound justification for omitting any correction for multiple comparisons, as it essentially allows an arbitrarily high rate of false positives.

One way the authors could validate the potential rate of false positives would be to shuffle the dendrite labels (i.e. randomly re-assign dendrites to each of the cell types) and re-run the analysis with the same statistical thresholds (i.e. p<0.02). Any genes that pass the significance threshold are by definition false positives, and the number of such genes can be compared with the number that are detected in the original (non-shuffled) analysis. This issue is critical because it affects the central claim of the paper that there are differentially expressed dendritic RNAs between cell types. As shown in Figure 3A, only ~2-30 such genes were detected even with the uncorrected p<.02 threshold, and it is thus conceivable that this number is strongly affected by false positives.

---

## [Author Response]

Reviewer #1:[…] It might be worthwhile for the authors to compare their data to a study published last year that carried out somatic vs. dendritic profiling in hippocampal neurons: Middleton, Eberwine and Kim, 2019.Although that study did not differentiate between neuronal cell types, presumably one could examine this using the raw data. It should probably be cited in this manuscript as well.

Thanks. We were aware of the above paper and cited it in the discussion of the original manuscript. We were initially hesitant to compare our datasets because (1) since only 16 neurons were profiled in their study there was insufficient power to perform the cell type identification and differential expression analyses that we used and (2) their study quantifies genes by number of reads instead of UMI counts (and most of the analyses we used have been designed for UMI counts). Nevertheless, we reasoned that an integration analysis as the one used in Figure 1F could circumvent the limitations of sample number and technical differences. As seen in Figure 1—figure supplement 2C, most of the dendritic and somatic samples from Middleton et al. 2019 integrated with our dendritic and somatic samples, respectively. This is consistent with our observation that dendrites and somata exhibit distinct transcriptomic signatures, and more diversity according to cell type is observed in somata. To fully reproduce our results, however, we believe the subcellular compartments of a similar or larger number of single neurons as the one used in this study, will be necessary.

While I am enthusiastic about the manuscript and the data it will provide to the field, there are some issues that could be addressed to improve the manuscript.1) In the very first figures the authors should describe how many dendrites are from which category of neurons or maybe highlight them with a different color in the UMAP plot (Figure 1C).

We have now added Figure 1—figure supplement 2A, which shows the number of dendrites and somata used in the clustering analysis of Figure 1C, as well as their subsequent cell type classification.

2) A potential deep comparison of the dendritic transcriptome of different cell types seemed exciting to me but was only superficially examined. The authors do mention the DEG list between the different groups, but only name a few genes and then validate them. It would be great to discuss the functional consequence of these differences in how the local dendritic transcriptome difference determines the functioning of these cells during signal transmission.

We agree this is a very interesting and relevant question which we tried to address using gene set enrichment analyses (GSEA). Author response image 1 shows the results of GSEA based on ranked by p-values from the various differential expression tests.

Only these gene ontology terms had an enrichment score of p-value < 0.05. Unfortunately, the above analyses likely underestimate the functional associations in the cell-type specific dendritic transcriptome due to several limitations. To avoid very noisy genes, we exclude genes expressed in less than 10% of samples and with fold change below 0.5 from the differential expression test. As such, we are only able to test 783 genes between Glutamatergic and GABAergic dendrites, 556 between GABAergic 1 and GABAergic 2/3 dendrites, and 753 between GABAergic 2 and GABAergic 3 dendrites. Then, in the GSEA analysis we only considered gene ontology terms with at least 10 members present in the differential expression results. This meant that of the 2302 possible gene sets we could only test enrichment of 237 gene sets in Glutamatergic vs. GABAergic results, 187 gene sets in GABAergic 1 vs. GABAergic 2/3 results, and 224 gene sets in GABAergic 2 vs. GABAergic 3. Future datasets with higher number of dendrites (as in Figure 5A, where 95 dendrites were considered), higher detection sensitivity and differential expression tests optimized for such datasets, will likely reveal a more complete picture of functional associations in the cell-type specific dendritic transcriptomes.

Also, when comparing and integrating the data with published tissue scRNA-seq, why not include the dendritic data too? Such comparisons could be important for others in the field to "deconvolute" their scRNA-seq data to look for dendritic-enriched transcripts.

We agree this an attractive possibility. Unfortunately, our data may not be fit for such analysis. Author response image 2 is an UMAP plot showing the integration of dendrites with somata and single cells from tissue.

**Author response image 2. respfig2:** 

Rather than integrate at particular locations, dendrites are distributed throughout multiple clusters. This is likely a reflection of their relatively shallower transcriptome (553 genes on average) when compared to somata (3,404 genes on average), which reduces their integration specificity and may also exaggerate the influence of a few variable genes.

Reviewer #2:This paper is excellent with a wealth of data. Some points to consider:1) In the first Results section they look at hippocampal cultures. The control is empty cuts. Could they perform an axon cut as a control?

Yes, this would be interesting but unfortunately the axons are just too thin to reliably obtain enough material for this kind of experiment.

2) They give the number of transcripts in the dendritic and somatic compartments. Although discussed in the Introduction and later in the manuscript, did they mention the total RNA extracted in each compartment and look for a selective loss of low abundance dendritic mRNAs normalized to total RNA in the beginning of the paper when making the simple comparisons to transcript numbers in the two compartments?

Thanks for this observation. We use the total number of counted RNA molecules (UMI counts) as a proxy for total RNA extracted (this includes both mRNAs and some non-coding RNAs) as shown in Figure 1B. In Author response image 3, we show the relation between expression and likelihood of detection (fraction of somatic or dendritic samples in which a gene is detected) in both compartments. There is a close relation between expression and detection in both somatic and dendritic samples. Thus, indeed, RNA species below a certain expression level are unlikely to be detected by our method.

**Author response image 3. respfig3:** 

a) It seems one way they addressed the above issue was by down sampling which is certainly reasonable. But could they determine whether dendritic mRNAs are selectively reduced in the mRNAs from the low abundance tail of somatic mRNAs? This analysis performed at some threshold would offer a simple and quick high level look.

This is also true. The lower a gene is expressed in the soma the less likely it is to be detected in dendrites as can be observed in Author response image 4.

**Author response image 4. respfig4:** 

3) The work superbly addresses quantitative comparisons-for example in the Poisson generalized linear model, but some greater to attention to functional differences would enhance the paper. This topic is not ignored-they have a section on "function associations" but that section is very descriptive and not completely as satisfying as one would expect for such profound differences and such an interesting problem. In this vein, their data would be very suitable for a bipartite community detection algorithm. It seems some of the points they want to make concerning dendritic and somatic mRNAs might become more revealing by identifying bipartite modules.

This is an interesting suggestion. We looked into this and see that the bipartite community detection algorithm has not been used much (yet) in these biological contexts. Unfortunately, we were not able to investigate this further during the revision period because of the substantial time investment required. We hope, however, to look into this in the future and of course the data are available for anyone who is interested to also analyze with this and any other algorithm.

4) They should address parvalbumin cells in some detail.

We do detect a subset of GABAergic 1 neurons that express *Pvalb*. This can be seen in Figure 1—figure supplement 3A (last line, second to last panel). These *Pvalb*+ subset of GABAergic 1 cells maps to the *Pvalb*+ Basket/Bistratified cells of the CA1 in Figure 1F. Thus, these cells would likely be separated into a separate cluster from the rest of GABAergic 1 neurons if more cells were to be added to our analysis. We now mention *Pvalb*+ neurons in the text describing Figure 1D and Figure 1F.

Reviewer #3:[…] Overall I found the paper to be well organized and clearly presented. Although the number of cells/dendrites and the number of transcripts per cell/dendrite are modest compared with scRNA-seq studies, due to the challenge of manual LCM, the data quality appears to be sufficient to clearly separate at least ~2 glutamatergic and 3 GABAergic cell types.1) The authors compare rat cultured neurons, derived from P0 animals, with large-scale scRNA-seq datasets from adult mouse primary hippocampus samples. What are the caveats from the species difference and also the difference between cultured and primary neurons? This should be at least mentioned.

Indeed, the integration performed here entails the comparison of datasets differing in multiple influential variables, namely sample source (culture vs. tissue), developmental stage (cells plated at P0 and cultured for 2 weeks vs. juvenile/adult brains) and species (rat vs. mouse). These variables could affect the integration in two not mutually exclusive ways: (1) differences in the transcriptomes of equivalent cell types, (2) differences in the repertoire of cell types. Integrating across different species has the additional caveat of differently named orthologs or missing orthology between species. The integration algorithm we used, Seurat v3, was shown to successfully integrate scRNA-seq datasets from different species, and from culture and tissue sources in (Butler et al., 2018), and from different developmental stages in (Mayer et al., 2018), and is now routinely used for the integration of this and many other kinds of single cell datasets (Stuart and Satija, 2019). To avoid errors due to differently named orthologs between rat and mouse, we converted rat gene names to the name of their mouse ortholog, when possible. These procedures do not exclude the possibility that the aforementioned variables contribute inaccuracies to the integration analysis. However, since our somata samples integrate to the other datasets in an organized manner consistent with the expression of established neuronal markers, we believe this integration is mainly driven by real biological similarities.

We now clearly mention these differences among datasets in the text describing Figure 1F and in the Materials and methods.

2) Data about gene expression are presented without any units (e.g. Figure 2A, B – y-axis; Figure 3A, B, Figure 4A-C). Are these values TPM, FPKM, CPM, or something else? In the axes with log units, is it showing log2 or log10?

Apologies. We quantify gene expression in the form of UMI counts, which are then processed to correct for differences in coverage and technical variation to obtained “Normalized UMI Counts”. We have now added this information to the graphs and to the Materials and methods. Unless otherwise specified we use natural logarithms.

3) In the differential expression analysis, the authors explain that they used an adjusted p-value (i.e. FDR control) for somatic cell type differences, but they chose an uncorrected p-value cutoff (p<.02) for the dendrites. The reasoning they provide is that the FDR-control is "overly punitive" (meaning conservative) and would cause a high rate of false negatives, which they demonstrate by downsampling the somatic datasets. This is not a statistically sound justification for omitting any correction for multiple comparisons, as it essentially allows an arbitrarily high rate of false positives.One way the authors could validate the potential rate of false positives would be to shuffle the dendrite labels (i.e. randomly re-assign dendrites to each of the cell types) and re-run the analysis with the same statistical thresholds (i.e. p<0.02). Any genes that pass the significance threshold are by definition false positives, and the number of such genes can be compared with the number that are detected in the original (non-shuffled) analysis. This issue is critical because it affects the central claim of the paper that there are differentially expressed dendritic RNAs between cell types. As shown in Figure 3A, only ~2-30 such genes were detected even with the uncorrected p<.02 threshold, and it is thus conceivable that this number is strongly affected by false positives.

Thanks for this suggestion. To address this concern, we used the suggested bootstrapping mock test approach. For each comparison we generated 1000 permutations where we randomly shuffled the identities of the dendrites between the two groups being compared and performed the same differential expression test in each. The reviewer was indeed right that a p-value of 0.02 contained a substantial number of false positives as seen in Author response image 5.

**Author response image 5. respfig5:** 

To correct for this, we reasoned that this approach could also be used to calculate an accurate and empirically-determined significance cutoff. We reasoned that the cumulative distribution of the permutation average in % can be interpreted as a false discovery rate. Thus, the p-value in which we obtain a 5% false discovery rate can be determined by the point in which we observe 5% of values in a cumulative distribution of the permutation average. Given the more rigorous approach we also felt we could increase the number of genes tested for differential expression by relaxing the required minimum fraction of samples expressing a gene from 0.19 to 0.1, and the required minimum log fold change between groups for a gene from 0.95 to 0.5.These analyses and the determination of p-values is now shown in Figure 2—figure supplement 1C, D, and described in the Materials and methods. Figure 3 has been updated accordingly.